# MiST: Understanding the Role of Mid-Stage Scientific Training in Developing Chemical Reasoning Models

## Abstract

Large Language Models acquire reasoning capabilities when fine-tuned in an online setting with simple rule-based rewards. Recent studies, however, indicate that success in this regard is conditioned on the latent solvability of tasks in the base LLM: RL can only amplify answers to which the base model already assigns non-negligible probabilities. This work investigates the emergence of chemical reasoning capabilities and what these prerequisites mean for chemistry. We identify two necessary conditions for RL-based chemical reasoning: 1) Symbolic competence, and 2) Latent domain knowledge. We propose MiST: a set of mid-stage training techniques to satisfy these, including data-mixing with SMILES-aware preprocessing and continued pre-training on a rich data mixture of 2.9B tokens. These steps raise the latent-solvability score on 3B and 7B models by 2x, and enable RL to lift top-1 accuracy from 10.9 to 63.9% on reaction name prediction, and from 40.6 to 67.4% on Conditional Material Generation. Similar results are observed for other challenging chemical tasks, while producing faithful reasoning traces. Our results define clear prerequisites for chemical reasoning training and highlight the broader role of mid-stage pre-training in unlocking reasoning capabilities.

## 1 Introduction

Reasoning tasks in chemistry are fundamental yet notoriously challenging, requiring models to integrate multiple layers of chemical knowledge and logical deduction (Coley et al., 2019; Alampara et al., 2024). While traditional chemoinformatics approaches rely primarily on supervised architectures optimized for specific tasks, they lack generalization and human-like reasoning capacities, instead often performing as highly specialized pattern recognition systems (Schwaller et al., 2019; Mirza et al., 2024a). Recently, reinforcement learning (RL) driven frameworks (Guo et al., 2025b; Narayanan et al., 2025; Zhao et al., 2025; Wang et al., 2025a) have shown promising advances in generating sophisticated emergent reasoning capabilities without explicit step-level supervision, achieving remarkable results across general-purpose domains like math and coding. Nevertheless, independent follow-ups have shown that such capabilities do not simply appear, but emerge instead as amplified patterns already existing in the base model's output distribution —even if with low likelihoods (Guo et al., 2023; Flam-Shepherd & Aspuru-Guzik, 2023). Consequently, whether RL succeeds on a new domain depends crucially on the latent solvability of the tasks for that specific base model.

Chemistry presents a severe stress test for this premise. Unlike arithmetic or programming, chemical problems combine highly specialized symbol systems (Weininger, 1988) (e.g. SMILES, IUPAC) with domain-specific physical constraints (valence, stereochemistry). Off-the-shelf LLMs typically fail to generate syntactically valid SMILES, let alone perform any tasks involving SMILES manipulation and generation (Bran et al., 2025). Empirically, we find that direct application of RL methods to such models fails: the reward signal vanishes because the correct answer never appears in the candidate set, except for the simpler examples.

These observations raise a fundamental question: What pre-training prerequisites must an LLM satisfy so that RL can reliably unlock chemical reasoning? In this paper, we answer that question by 1) proposing quantitative diagnostics that measure a model's latent solvability for chemical tasks, 2)

systematically creating and ablating two proposed domain-specific prerequisites, and 3) showing that RL and other reasoning post-training techniques succeed if those diagnostics cross certain thresholds.

We propose symbolic competence and latent chemical knowledge as two necessary prerequisites for reasoning in chemistry. The former requires that models must be able to read and generate syntactically valid chemical strings, like SMILES, IUPAC names, or CIF files. The second requirement means that answers must exist in the long-tail of the model's prior distribution, so that these can be exploited by the RL training. We demonstrate, through a diagnostic benchmark for latent solvability, that improving on these requisites boosts model post-RL performance by up to 29%, yielding highly capable chemical reasoning models.

We perform a range of ablations and generalization tests on RL performance across an array of tasks, and show that removing any single prerequisite collapses RL gains, confirming their necessity. Our findings provide a framework to inform how reasoning-oriented RL methods for LLMs will perform when applied to complex scientific domains. It serves effectively as a predictive and selective framework to optimize against, before any expensive RL is performed.

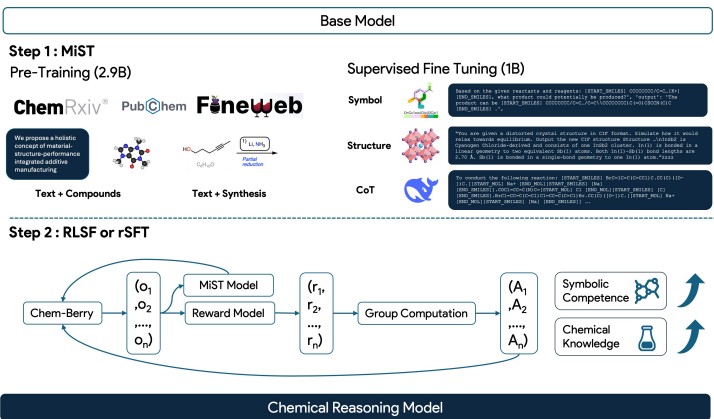

Figure 1: Multi-stage pipeline for training a chemical-reasoning language model. Step1 (*MiST*, 3.9 B tokens) Continued Pretraining exposes a general-purpose base model to a chemistry-centric corpus that interleaves plain text with compound & synthesis information. A subsequent 1 B-token supervised fine-tuning phase teaches three formats: (i) symbol-level molecular or material understanding, (ii) structure-aware question & answers, and (iii) chemical chain-of-thought (CoT). In Step2 the MiST backbone is further specialized with either RLSF (reinforcement learning from scientific feedback) or rSFT (reasoning-style supervised fine-tuning). A pool of candidate answers $(o_1, \ldots, o_n)$ generated by the MiST model is scored by a task-specific reward model $(r_1, \ldots, r_n)$; a group-computation module aggregates these signals to update the policy, iteratively refining the model into a *Chemical Reasoning Model*.

## 2 RELATED WORK

**Post-training methods for reasoning**    The standard recipe for aligning LLMs augments supervised fine-tuning (SFT) with reinforcement learning from human or synthetic feedback (RLHF/RLAIF) (Lee et al., 2024). While RLHF reliably improves helpfulness and stylistic alignment, it is often insufficient for multi-step reasoning. Subsequent work therefore introduced chain-of-thought distillation (Wei et al., 2022; Li et al., 2023), and tree search with self-consistency (Xie et al., 2024b). A recent, influential result by Guo et al. (2025b) showed that *rule-based* rewards can unlock strong mathematical and coding skills, provided that the base model already allocates non-negligible probability mass to correct answers. Independent analyses confirmed that RL mainly acts as an *amplifier*: it can only surface solutions that lie somewhere in the base distribution (Yue et al., 2025). For weaker bases, SFT on traces generated by a larger model often outperforms RL (Guo et al., 2025b). Our work adopts this "RL as amplifier" view and asks what pre-training conditions make chemical problems *latently solvable* so that RL can succeed.

**Chemical language modeling** Language models have been adapted and used for a range of chemical tasks (Caldas Ramos et al., 2025). These models typically operate on linearized molecular strings, such as SMILES (Weininger, 1988), SELFIES (Krenn et al., 2020), or IUPAC names. Masked pre-training approaches (ChemBERTa (Chithrananda et al., 2020), MolBERT (Fabian et al., 2020)) learn molecular fingerprints that are useful for QSAR, whereas the Molecular Transformer family targets forward and retrosynthesis prediction (Schwaller et al., 2019; 2020). More recently, LLMs have been adapted and applied for tackling chemical tasks (Frey et al., 2023; Zhang et al., 2024; Jablonka et al., 2024; Xie et al., 2023b); extending general LLMs for molecule generation, property prediction, and Q&A. Other works have adapted LLMs for use as chemistry agents, integrating robotic labs and other tools (Bran et al., 2023; Boiko et al., 2023), hypothesis generation (Yang et al., 2025), and more recently, workflows have been designed for molecular design and synthesis planning (Wang et al., 2024a; Bran et al., 2025).

**Mid-stage domain adaptation** Continuing pre-training on an in-domain corpus—often called *domain-adaptive pre-training* (DAPT) or *continued pre-training* (CPT)—has become the dominant recipe for turning a general LLM into a domain specialist. Early successes such as BioMegatron for biomedicine (Shin et al., 2020), Legal-BERT (Chalkidis et al., 2020), and Code-Llama for programming (Rozière et al., 2023) demonstrated sizable gains with only a few billion extra tokens. A recent wave of work scales the idea to scientific domains: (1) AdaptLLM (Cheng et al., 2023) shows that a 7B parameters model, after just 10–15B of financial tokens, rivals BloombergGPT (50B parameters) on in-domain QA; (2) Tag-LLM (Shen et al., 2024) and Efficient-CPT (Xie et al., 2024a) report similar jumps while using parameter-efficient adapters; (3) SciLitLLM (Li et al., 2024b) uses a 12.7B token corpus of textbooks and full-text papers and beats much larger baselines on scientific-literature understanding; (4) domain-specific studies in materials science (Lu et al., 2025), radiation oncology (Holmes et al., 2023), Japanese finance (Hirano & Imajo, 2024), and cybersecurity (Bayer et al., 2024) confirm that CPT injects *latent* domain knowledge that survives further instruction tuning. However, two caveats emerge: CPT can erode zero-shot prompting ability if done naively (Cheng et al., 2023), and very small models ($< 2B$ parameters) often fail to develop new capabilities even after extensive CPT (Lu et al., 2025; Hsieh, 2025). Crucially, none of these works evaluate whether the adapted model becomes *latently solvable* for multi-step reasoning tasks that reinforcement learning could later amplify.

The chemical domain remains comparatively under-explored. ChemBERTa-2 (Maziarka et al., 2023) continues a BERT-style encoder on ∼1B SMILES tokens and improves fingerprint-style QSAR, while ChemLLM (Zhang et al., 2024) , DARWIN-Chem (Xie et al., 2023a), and SciDFM (Sun et al., 2024) incorporate reaction patents or literature but still operate in a single-shot, pattern-recognition regime.

**LLM capability diagnostics** Benchmark accuracy and perplexity offer only coarse snapshots of a model; they ignore the richer signal contained in the full conditional probability distribution. Holistic evaluation suites such as HELM (Liang et al., 2022) and LiveBench (White et al., 2024) log likelihoods but still aggregate them into single numbers. Probability-based *intrinsic* probes provide finer insight. BLiMP minimal pairs measure grammatical preference gaps (Meister & Cotterell, 2021), an idea later reused to analyze in-context learning brittleness (Zhao et al., 2024) and out-of-domain (OOD) intent detection (Wang et al., 2024b). For factual QA, calibration studies show that token-level probabilities reveal when models "know what they know" (Jiang et al., 2021; Kadavath et al., 2022). Distributional uncertainty metrics now underpin OOD detection (Liu et al., 2024a), self-correction pipelines (Liu et al., 2024b), and medical-reasoning assessment (Li et al., 2024a). Pezeshkpour (2023) and Wang et al. (2024c) formalize diagnostics as distribution-matching problems using KL divergence or Wasserstein distance, while (Ye et al., 2024) links dispersion measures to downstream robustness.

## 3 PRELIMINARIES

We formalize the notions used throughout the paper and introduce the metrics that constitute our diagnostic suite.

## 3.1 Prerequisite 1: Symbolic Competence

To assess the symbolic competence of models, we compute the likelihood of generating a given sequence, in our case, a set of SMILES strings. We use a dataset of 10,000 molecules obtained from PubChem (Kim et al., 2025), and use the following definitions to compute a symbolic competence score.

**Token log-likelihood extraction**    Given a model $p_\theta$ and a SMILES string $s = (t_1, ..., t_L)$.

At position $i$ we compute the log-likelihood $r_{i,p_\theta}(s)$ of ground-truth token $s_i$ within $p_\theta$'s next-token distribution $r_{i,p_\theta}(s) := p_\theta(t_i|t_1...t_{i-1})$. The mean of the whole string is taken as in eq. 1.

$$r_{p_\theta}(s) = \frac{1}{L} \sum_{i=1}^{L} r_{i,p_\theta}(s) \qquad (1)$$

**Symbolic competence score**    We define the symbolic competence score (SCS) on the assumption that a symbolically competent model should assign better likelihoods to chemically correct strings than to corrupted or invalid ones. We therefore measure the separation in the distributions of mean ranks between valid (canonical) SMILES and corrupted ones (eq. 2).

where $\sigma_1$ and $\sigma_2$ are each set's standard deviations, and $\sigma_{pool}$ is the pooled standard deviation of the two sets. SCS is the Cohen's d effect size, where higher values indicate a cleaner separation and therefore stronger symbolic competence. A score of 0 means the model cannot distinguish canonical from corrupted strings, while SCS $\approx 2$ corresponds to > 95 % separation. $\mathcal{C}$ is

$$SCS := \frac{\bar{r}(\rho(m)) - \bar{r}(\mathcal{C}(m))}{\sigma_{pool}}, \qquad (2)$$

$$\sigma_{\text{pool}} = \sqrt{\frac{(n_1 - 1)\sigma_1^2 + (n_2 - 1)\sigma_2^2}{n_1 + n_2 - 2}} \qquad (3)$$

a SMILES canonicalization operator, while $\rho$ is a SMILES corruption operator that randomly deletes grammar characters with some probability (here chosen to be 0.2 —see Section 5.1), effectively yielding invalid but similar SMILES. For the Conditional Material Generation task, instead of corrupting and calculating the SCS on SMILES, the calculations are performed on compositions, which specify their elements and space group in the format: A B A B <sg*X*>, where A and B are elements, and *X* represents the space group number.

## 3.2 Prerequisite 2: Latent Chemical Knowledge

As has recently been shown, the role of RL in training reasoning LLMs seems to be that of an amplifier, i.e., correct answers already exist in the base model's prior distribution with non-negligible probability.

With this in mind, we aim to assess the latent chemical knowledge of a given base model. As a proxy to this, we adopt the same strategy as that we use with the symbolic data, by measuring the Chemical-Competence Score (CCS), defined as the difference in the distributions of mean ranks between factually correct chemical statements and wrong ones. Given a list of chemical statements, such as the SMolInstruct Molecule Description subset (Yu et al., 2024b), we generate corrupted data by randomly swapping one sentence from each original statement with that from another randomly chosen statement in the pool.

## 3.3 Post-training methods

Large-scale pre-training furnishes the *prerequisites* discussed in Section 3.1–3.2. We now describe the two post-training methods that we use throughout this work to bake-in and amplify these capabilities.

**Supervised fine-tuning on reasoning traces**    (Guo et al., 2025a) showed that small base models can be trained with SFT on reasoning traces, resulting in small reasoning models that mimic the behavior demonstrated in the SFT training data, even if such data does not directly target the specific downstream task the models are evaluated on. The reason is that SFT transfers the response style and not only the task-specific capabilities, thus serving as an *amplifier* of latent knowledge. Following this, some reasoning traces were distilled from DeepSeek-R1 and used to perform SFT on our

pretrained models. We generated $\sim 600{,}000$ solutions for two canonical tasks: IUPAC $\rightarrow$ SMILES and SMILES $\rightarrow$ IUPAC, based on PubChem compounds.

**Reinforcement learning with verifiable rewards**  Following recent works (Wang et al., 2025b), we adopt Reinforcement Learning with Verifiable Rewards (RLVR) as a post-training method for our models. In this context, models are trained online with rule-based rewards that depend entirely on the final outcome. The goal of this type of training, as exemplified in previous works (Wang et al., 2025b) is to encourage the model to achieve good results on the training tasks, while developing intermediate strategies to achieve this, that might involve reasoning.

We designed and used different types of reward functions for our GRPO experiments: (1) formatting rewards to ensure separation between the model reasoning and answer, (2) accuracy rewards to verify the correctness of the model answer, (3) helper rewards to penalize the model if the completions are ill-formed (such as very short completions, repetitive behaviors etc.). For the accuracy rewards, we employed different approaches to compare the answer and the solution, such as exact matches, Tanimoto similarity between SMILES, or Levenshtein, see Appendix D.2.1.

**Downstream reasoning tasks**  To train and evaluate the reasoning capabilities of our models, we implemented a suite of challenging tasks relevant to chemistry. The tasks have been selected with the following criteria in mind: (1) Difficulty: the task must be challenging enough to be unsolvable by base models alone, (2) Reasoning-suitable: tasks must be suitable for reasoning, i.e. solving an instance of the task would require more System-2 thinking from human experts than System-1, and (3) Dataset availability: Datasets must be readily available such that, upon adaptation, an input-outcome dataset can be built that is representative of the task. The final list of tasks is listed in Table 2, and implementation details are provided in the Appendix B.

## 4 MɪST: Mɪᴅ-ꜱᴛᴀɢᴇ Sᴄɪᴇɴᴛɪꜰɪᴄ Tʀᴀɪɴɪɴɢ

The purpose of this mid-stage training is to enhance the model's ability to generate valid SMILES, accurately follow chemistry-focused instructions, and strengthen its general chemical knowledge. We do this by continuing pretraining (next token prediction objective) on chemical and SMILES-related data, and then by performing SFT to improve instruction-following and increase the context window length.

### 4.1 Dᴀᴛᴀꜱᴇᴛꜱ

To construct the pretraining data, we used the data mixture as described in Table 1, obtained and processed as described in Appendix E. All the data underwent the same preprocessing pipeline to interleave SMILES with text whenever a molecule name appeared (e.g. IUPAC, common name, short form, etc), similar to (Taylor et al., 2022). We additionally generated a synthetic dataset using RDkit (RDKit, online) extracted properties of molecules (like QED, TPSA, etc) and filled it in a template. Furthermore, we include a "replay" dataset aiming to preserve the model's natural language abilities while furthering it's learning about chemical knowledge. We chose the Qwen2.5-3B base model to perform the pretraining for 3 epochs.

For SFT, as shown in Table 1 we used question-answering (QA) training examples derived from SmolInstruct (Yu et al., 2024a), specifically employing only the SMILES↔IUPAC and molecule captioning subsets. We also collect examples from MPtrj dataset(Deng et al., 2023). Additionally, we incorporated MMLU and chain-of-thought (CoT) reasoning traces from DeepSeek-R1, which were preprocessed to maintain coherence with our pretraining data. In this phase, we also expanded the model's context window from 4,096 to 8,192 tokens to accommodate longer reasoning sequences. The pretrained model underwent SFT for approximately 8 epochs, continuing until the previously observed loss spikes were fully mitigated.

Table 1: MiST training data recipe. CP = continued pretraining; SFT = supervised fine-tuning.

| Stage | Dataset Source | Tokens / Samples | Percent (%) |
|---|---|---|---|
| **CP** | ChemRxiv + S2ORC | `1.2B` | 41.38 |
| | FineWeb (chemistry-filtered) | `1.4B` | 48.28 |
| | PubChem synthetic (600k compounds) | `220M` | 7.59 |
| | CommonCrawl Replay | `80M` | 2.75 |
| | **Total (CP)** | **$\sim$`2.9B`** | **100** |
| **SFT** | DeepSeek reaction traces | $\sim$7K samples | 0.22 |
| | DeepSeek relaxation traces | $\sim$2K samples | |
| | MPtrj dataset | $\sim$20K samples | 0.65 |
| | SmolInstruct | >3M samples | 98 |
| | MMLU | Train $\sim$350; Chemistry $\sim$300 | |
| | CoT Chain | $\sim$27K samples | 0.88 |
| | **Total (SFT)** | **$\sim$`1B`** | **100** |

## 5 RESULTS

As an initial downstream test of our pipeline's performance, we use ChemBench (Mirza et al., 2024b) to evaluate the general chemistry knowledge of LLMs; the results are shown in Table 2. This evaluation is a first check to ensure our pipeline consistently improves performance on an important public benchmark for chemistry, before we begin studying the downstream effects of MiST on RL training. The results show an overall +10% improvement in performance bringing it up to 45% which, better than GPT-3.5-Turbo and Llama-3.1-8B-Instruct Mirza et al. (2024a), models at least 2x larger than our resulting 3B model. The role of MiST is particularly important in Organic, Inorganic, and General Chemistry, with improvements of up to 6-7% over the Qwen+SFT baseline, and more than 11% over the base (instruction-tuned) model, suggesting benefits of both post-training stages on model's chemical performance. These results serve already as diagnostic measures of success of a given mid-training methodology, and serve as a basis to select models for the following stage of RLVR, with the goal of enhancing reasoning and problem-solving capabilities.

We then proceed to evaluate model's capacity of learning in an online setting from verifiable rewards through RLVR. As explained in Section B, we implement a number of chemical tasks that are suitable for reasoning, and for which verifiable rewards can be defined (see Appendix B). Several models were trained on this setting and, in the following, we evaluate task performance as a function of the base model used. We employ two different inference techniques to generate the results reported here, namely using System-1 (direct answer) and System-2 (employing reasoning) thinking, as defined by (McGlynn, 2014). We do this by appending the tags "<answer>" or "<reasoning>" respectively upon

Figure 2: ChemBench sub-domain Accuracy (%). Results obtained based on Qwen-2.5-3B

| | Models | | |
|---|---|---|---|
| Sub-domain | Qwen Inst | +SFT | +MiST (ours) |
| Organic | 44.99 | 46.15 | **50.12** |
| Inorganic | 46.70 | 51.08 | **57.60** |
| Tox/Safety | 21.33 | **26.52** | 26.37 |
| Material Sci | 35.84 | 42.50 | **48.75** |
| General | 33.56 | 38.25 | **44.30** |
| Preference | 45.40 | 50.00 | **52.10** |
| Analytical | 25.00 | 34.20 | **40.70** |
| Technical | 42.11 | 44.74 | **50.00** |
| Physical | 20.60 | 35.10 | **38.78** |
| Total | 35.06 | 40.95 | **45.41** |

generation, which induces models into either type of thinking. The results shown in Table 2 show the performances of our models across the multiple tasks defined in Section B, along with the diagnostic metrics defined in Section 3.1.

The results reveal the large effect that the MiST proposed here has on symbolic competence, as demonstrated by the SCS column. Clearly pretrained models like Qwen2.5-3B lack the symbolic abilities needed to complete tasks requiring SMILES understanding and writing. However, this is overcome with MiST. Furthermore, the results show that RL generally improves the performance of

Table 2: **Effect of MiST and each post-training stage on downstream reasoning tasks.** SCS = symbolic-competence score, CCS = chemical-competence score; both are unitless effect-size measures ranging from 0 (no separation) to 2 (near-perfect separation); higher is better, see Section 3.1. I2S = IUPAC→SMILES translation, RxP = forward reaction prediction, RxN = reaction-naming, CMG = conditional material generation. For the three downstream tasks we report top-1 accuracy. The value outside the parentheses is obtained with a "direct answer" (system-1) prompt. Values inside parentheses are the accuracy when "reasoning" (system-2 chain-of-thought) prompting is induced.

| | Metrics | | Reasoning tasks | | | |
|---|---|---|---|---|---|---|
| Model | SCS ↑ | CCS ↑ | I2S ↑ | RxP ↑ | RxN ↑ | CMG ↑ |
| **Qwen-2.5 3B** | 0.955 | 0.352 | 0.0 | 0.0 | 10.33 (10.9) | 40.6 (0.5) |
| +CP | 1.561 | 0.404 | 1.0 | 0.0 | 11.1 (10.3) | 40.1 (3.9) |
| +SFT | 1.650 | 0.707 | **52.7** | 5.2 (13.6) | 15.1 (17.5) | 38.9 (4.0) |
| +RL(I2S) | 1.535 | 0.695 | 52.0 | 3.2 (12.2) | 13.3 (17.2) | 42.1 (39.8) |
| +RL(RxP) | 1.880 | 0.782 | 49.9 | **6.6 (17.4)** | 14.5 (16.9) | — |
| +RL(RxN) | 1.650 | 0.698 | 48.9 | 5.8 (9.0) | **28.5 (46.8)** | 43.6 (15.2) |
| +RL(CMG) | 0.119 | 1.737 | 50.4 | 0.0 (7.8) | 13.1 (18.2) | **64.9 (70.5)** |
| **Qwen-2.5 7B** | 0.97 | 0.406 | 0.0 | 0.0 (0.2) | 10.7 (12.1) | 40.8 (36.3) |
| +CP | 1.67 | 0.445 | 0.2 | 0.8 (0.4) | 14.8 (14.7) | 45.6 (33.8) |
| +SFT | 1.74 | 0.775 | **65.7** | 13.2 (25.2) | 13.8 (30.1) | 38.2 (5.5) |
| +RL (I2S) | 1.67 | 0.766 | 65.2 | 12.6 (25.2) | 22.7 (31.4) | 38.5 (30.6) |
| +RL (RxP) | 1.71 | 0.770 | 65.2 | **15.6 (29.8)** | 11.7 (31.2) | 39.6 (18.2) |
| +RL (RxN) | 1.73 | 0.731 | 61.7 | 13.2 (12.6) | **26.4 (63.9)** | 23.6 (37.7) |
| **MiST Ablations (7B)** | | | | | | |
| no MiST + RL (RxP) | 1.03 | 0.408 | 0.0 | 0.0 (0.0) | 9.97 (12.1) | — (—) |
| **Baselines** | | | | | | |
| ChemLLM-7B | 1.18 | — | 0.5 | 2.04 (—) | 18.7 (—) | — (—) |

LLMs on specific chemical tasks, and this effect is remarkably stronger on tasks requiring SMILES synthesis, like *Reaction Prediction* or *Iupac 2 SMILES*.

One important observation from these results is that activating reasoning on RL-trained LLMs generally yields better results; however in certain cases this trend reverses, as is the case of *Iupac 2 SMILES*. In this task, we measure better performance when reasoning is not activated, however the gap is smaller for the RL-trained models. We attribute this to the ability already being present *and* amplified in the LLM after SFT, which during RL training hinders learning of different task-solving patterns as models already perform well at that stage. Further research should go into this direction. Similar effects are observed across model scales (3B and 7B), indicating that these results can further generalize to larger model sizes to yield much improved chemical reasoning models.

Previous works have built models for similar tasks Zhang et al. (2024); Narayanan et al. (2025); Xia et al. (2025). In Table 2, we compare against a baseline of similar size, ChemLLM, a model trained starting from InternLM2-Base-7B Cai et al. (2024). Our results show that, on the tasks evaluated, ChemLLM is outperformed by our RL-tuned 3B models. The advantage is especially striking in the tasks of reaction naming and and reaction prediction. This contrasts with the results presented in the original publication, reporting over 88% 5-shot accuracy on retrosynthesis which could not be reproduced.

## 5.1 FORMULATION OF SCS

The Semantic Competence Score (SCS) as defined in 2 depends critically on **a.** a suitable dataset, and **b.** a corruption rate. As discussed, the dataset has been selected on the basis of what constitutes a suitable dataset for the tasks at hand. For the organic chemistry tasks (RxP, RxN, etc) we have selected a dataset of molecules in SMILES format, which represents a distribution of fully semantically correct instances. The corruption rate is then an artifact used to degrade that distribution into one of non-semantically-valid instances.

The SCS is computed on the basis of a model being able to distinguish between the two distributions, as measured by Cohen's d effect size (eq. 2). In this work the corruption rate has been determined empirically aiming to balance enough resolution power (high enough corruption), without making it too obvious a task with an entirely random distribution. Figure 3 shows this for both Qwen-3B and 7B variants, across different MiST and RL treatments and measured on different corrupt rates. As can be seen, at cr=0.2 models are already able to distinguish suitably between distributions, and there is a clear gap between base models and models treated with MiST. As cr increases, the gap remains but it tends to decrease, especially in the case of Qwen-7B, indicating that at too high corruption rates, even small base models can correctly distinguish corrupt from non-corrupt smiles. At too low cr (0.1), the gap is also smaller.

## 5.2 SCS AS A PREDICTIVE FRAMEWORK

A main claim of this paper is that it is important to develop diagnostic metrics to help prognosticate the performance of models before RL is applied, similar to the role of scaling laws in pre-training Kaplan et al. (2020). Here we show that pre-RL SCS is effective for this. As shown in Figure 4, pre-RL SCS reliably predicts post-RL success of models. It is observed that, on the tasks evaluated, low pre-RL SCS predictably yields incapable models, while high SCS leads to better models, especially when the models are trained with RL on the tasks being

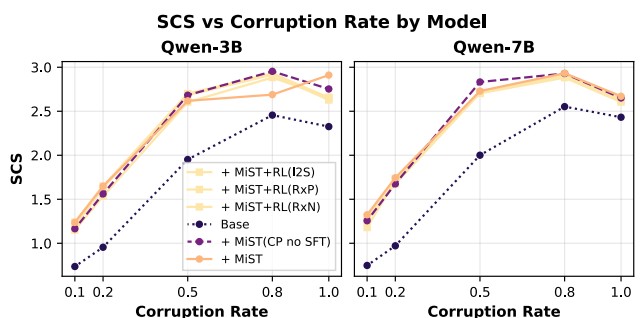

Figure 3: Selection of SCS values.

evaluated. We find strong correlations between pre-RL SCS and post-RL performance, namely $\rho = 0.64$ for reaction prediction and $\rho = 0.60$ for IUPAC translation across both 3B and 7B models. This provides a quantitative empirical framework to support future model development: SCS > 1.5 is a necessary threshold for RL to unlock reasoning.

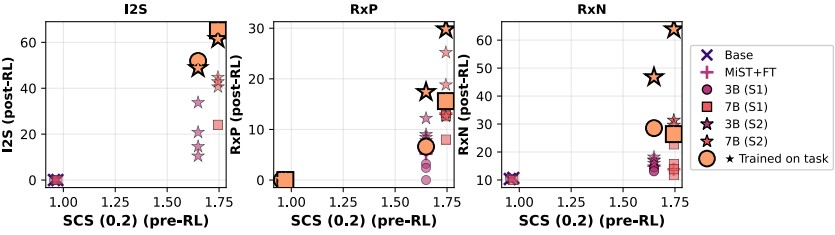

Figure 4: **Post-RL task performance vs. pre-RL SCS(0.2)**. Markers indicate model size (circles = 3B, squares = 7B) and system type (filled = S1, stars = S2). Reference markers: 'x' = base Qwen, '+' = MiST+FT. Red markers with '⋆' indicate the model trained specifically on that task. Colors: darker = 3B, brighter = 7B (magma palette).

As case-study, we use SCS to retrospectively select the best base model to train ether0, the first chemical reasoning model Narayanan et al. (2025). We selected a range of open-source base LLMs across a variety of providers and sizes between 14B and 70B, and measured SCS. The results in Table 5 show that Mistral-24B would be the best option, followed by Gemma-2-27B, while the rest of models, even some beyond 70B in size, fall below the 1.5 SCS

Figure 5: SCS across larger base LLMs.

| Model | SCS |
|---|---|
| Qwen-2.5-14B Qwen et al. (2025) | 1.161 |
| Qwen-2.5-32B Qwen et al. (2025) | 1.238 |
| Qwen-2.5-72B Qwen et al. (2025) | 1.094 |
| Llama3-70B Grattafiori et al. (2024) | 1.481 |
| Gemma-2-27B Team et al. (2024) | **2.097** |
| Mistral-24B Mistral AI Team (2025) | **2.324** |
| ether0 (Mistral-24B) | 1.597 |

limit established previously. Within a similar size range, and despite its much better scores at most public benchmarks Mistral AI Team (2025), Qwen-2.5-32B is predicted to be a much worse base model for RL in chemistry tasks. These results match the base model selection from ether0 Narayanan et al. (2025), that effectively used Mistral-24B as a base, further increasing confidence on the predictive power of measures of this type.

## 6 DISCUSSION

This paper set out to answer a concrete, practical question: What conditions must a general-purpose LLM satisfy so that light-weight, rule-based post-training methods (SFT + RLVR) can unlock reliable chemical reasoning? We conducted a series of experiments using the Qwen2.5-3B and 7B models as bases. Our results suggest that a key requisite, Symbolic Competence on the base models (pre-RL), can predict post-RL success. We demonstrate this result by proposing Mid-stage Scientific Training (MiST), which we show 1. increases SCS on base models, and 2. predictably produces models that, under the same RL training, outperform non-MiST baselines. We take a step further and retrospectively analyze base model choices behind recent releases of reasoning models. We find that, out of the most relevant open-source base models between 14B and 70B, Mistral-24B stands out as the one with the highest SCS, which retrospectively matches the choice for ether0.

Furthermore, as already indicated in other recent reports, RL remains an amplifier of already existing behaviors and knowledge in base LLMs. However more importantly for the case of chemistry, and other scientific fields that make heavy use of domain-specific terminology and symbolic systems, symbolic competence remains the true bottleneck, at least for small-scale LLMs. As our results demonstrate on SMILES-heavy tasks, like *Reaction Prediction* or *IUPAC to SMILES*, base models barely perform on these tasks, with results nearing 0% accuracy. SFT only boosts the Reaction Prediction results to 5.10%, however MiST is necessary to boost accuracies to 25.2% when reasoning is activated, also shows the improvements in the material science knowledge that not specifically trained (in CMG task),indicating a strong role of MiST in enabling the solution of scientific tasks.

Our findings generalize beyond chemistry. Any scientific field that (1) relies on specialized symbol systems and (2) has access to verifiable rewards can likely benefit from the same two-stage recipe: (1) ensure symbol mastery via targeted MiST, (2) apply RLVR or other post-training techniques to amplify latent solutions. With this we show that, small, compute-efficient models can already reach useful competence if those prerequisites are met. MiST demonstrates that carefully crafted, mid-stage scientific training is a powerful lever for unlocking reasoning in specialized domains. Rather than chasing ever larger parameter counts, we advocate investing in domain-specific data pipelines and intrinsic diagnostics—ingredients that, as chemistry shows, can turn an otherwise myopic LLM into a competent scientific assistant.

## 7 LIMITATIONS

While MiST demonstrates that targeted mid-stage pre-training can unlock chemical reasoning in a 3B-parameter model, several caveats remain. First, the SCS criterion works especially well in domains and tasks where the outcome validity is verifiable and easily corruptible, as is the case with chemistry and SMILES notation. We find such complications in the case of CMG, where CIF files are less-trivially corruptible, leading to worsened predictive power. Using a 100% valid notation, as is the case of SELFIES Krenn et al. (2022) for molecule generation tasks, can also compromise the value of SCS-like measures. Similar metrics can however be devised for biological sequential data, mathematical notation, etc. Second, the RLVR rewards we use focus on syntactic agreement with ground truth (e.g. exact SMILES or high Tanimoto similarity) and thus do not discourage chemically implausible or unsafe outputs, leaving open the possibility of reward hacking. Third, our evaluation suite—reaction prediction, IUPAC to SMILES translation, and conditional material generation, cover a narrow slice of chemistry; tasks that hinge on stereochemistry, kinetics, spectroscopy, or three-dimensional conformations remain unexplored. Finally, our pre-training corpus is dominated by small-molecule, organic literature and patents, potentially biasing the model against inorganic, macromolecular, or bio-chemical domains. Addressing these limitations will be critical before SCS can be routinely used as a diagnostic metrics in other domains, and for MiST-style models to be relied upon as general scientific reasoning models.

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

## A  APPENDIX

## B  CHEMICAL TASKS

This section compiles all of the chemical tasks we have implemented for training models with RLVR, including details of the dataset compilation, rewards used, along with additional experimental results not shown in the main manuscript. The details for each task are specified with a table containing a summary of the rewards used, short explanations of what the input and expected outputs are, and finally, some sample outputs from the trained models. Following the table, we report additional experimental results, implementation details, and other task-specific details.

### B.1  REACTION PREDICTION (RxP)

---

***Task: Reaction prediction***

---

| | |
|---|---|
| Dataset Source | USPTO-480K (Coley et al., 2019) |
| Input | `Reason and predict the correct product in SMILES notation from the following reaction: CCN(CC)CC.COc1ccc(O)cc1.ClC(Cl)Cl.ClCCl.O.O=C(Cl)c1ccccc1.` |
| Target Molecule | `COc1ccc(OC(=O)c2ccccc2)cc1` |

- - - - - - - - - - - - - - - - - - - - - - - - - - - - - - - - - - - - - - -

| | |
|---|---|
| Output | <think> The reaction involves the acylation of a phenolic hydroxyl group using benzoyl chloride `O=C(Cl)c1ccccc1` in the presence of triethylamine `CCN(CC)CC` as a base. Here's the step-by-step mechanism: ...
  1. Reactants Identification**: ...
  2. Mechanism**: ...
  3. Product**: ...
 Final Answer**: The product SMILES is `COc1ccc(OC(=O)c2ccccc2)cc1` representing the acetylation of the phenolic oxygen [O] in 4-methoxyphenol `COc1ccc(O)cc1` by benzoyl chloride `O=C(Cl)c1ccccc1`, facilitated by triethylamine `CCN(CC)CC`. </think>
 <answer> `COc1ccc(OC(=O)c2ccccc2)cc1` </answer> |

- - - - - - - - - - - - - - - - - - - - - - - - - - - - - - - - - - - - - - -

| | |
|---|---|
| Answer | `COc1ccc(OC(=O)c2ccccc2)cc1` ✓ |

---

Table 3: Example of reaction prediction task.

The reaction prediction task requires the model to reason and predict the correct product molecule given a list of reactants and reagents (Table 3). Solving this task usually requires expert chemists to think about the reactivity of the reactants involved, and propose and evaluate different reaction mechanism hypotheses. These serve as arguments and causal explanations that support the decisions.

The dataset for the RLVR training of this task was derived from the USPTO-480K (Coley et al., 2019) after removing the samples used in the SFT phase. 50K reactions were randomly chosen for the training set, and 500 reactions for the test set.

Given a model output $o$, from which a final answer $a$ can be extracted, the reward function is the sum of format correctness ($R_{\text{format}} : o \mapsto [-1, 1]$, see Appendix D) and accuracy of the predicted product ($R_{\text{acc}} : a \mapsto \{-1, -0.5, 1\}$). The accuracy reward is determined by an exact match check against the ground truth:

$$R_{\text{acc}}(a) = \begin{cases} -1, & \text{if Ans cannot be captured from Output or is not a valid SMILES.} \\ -0.5, & \text{if Ans refers to a molecule different than the ground truth.} \\ +1, & \text{if Ans corresponds to the ground truth molecule.} \end{cases}$$

Figure 6 illustrates the evolution of the accuracy reward throughout training. The base Qwen2.5-3B model plateaus early at a reward below the -0.5 threshold, indicating that while it frequently generates

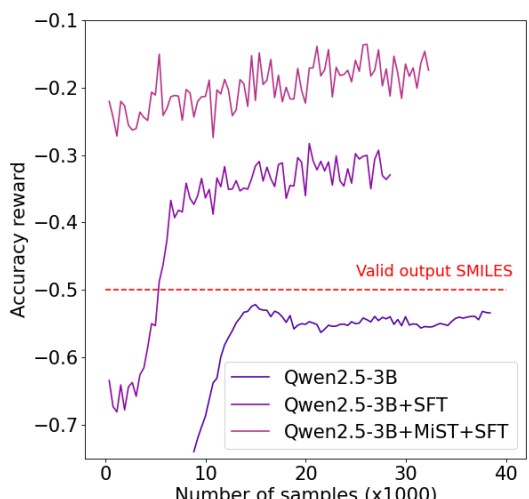

Figure 6: Accuracy reward evolution.

syntactically valid SMILES strings, it fails to predict the correct product molecules. In contrast, both fine-tuned variants (Qwen2.5-3B+SFT and Qwen2.5-3B+MiST+SFT) maintain accuracy rewards above -0.5 in the majority of the training process. The SFT-only model shows a sharp increase in reward during the first 8,000 training samples, reflecting rapid format alignment. This is followed by a more gradual rise, suggesting improved chemical reasoning over time. Notably, the model pretrained with MiST before SFT follows a similar trajectory but consistently achieves a higher reward ceiling. This highlights the effectiveness of MiST and SFT in enhancing the model's learning capability during RLVF.

## B.2 IUPAC TO SMILES (I2S)

Table 4: Example of IUPAC→SMILES conversion.

**_Task: IUPAC to SMILES Conversion_**

| | |
|---|---|
| Dataset Source | PubChem (CID 2,000,001 to 3,000,000) (Kim et al., 2025) |
| Input | `Question:  You are an expert in Cheminformatics, who is very familiar with Simplified Molecular Input Line Entry System (SMILES) notation, and here's a task for you:  Given a molecule with the IUPAC name as below, please provide the corresponding SMILES notation.  Here is the IUPAC name:  "4-(azepan-1-ylsulfonyl) -N- (pyridin-3-ylmethylideneamino) benzamide".` |
| Target Molecule | `O=C(NN=CC1=CC=CN=C1)C1=CC=C(S(=O)(=O)N2CCCCCC2)C=C1` |
| Output | `<think> The SMILES representation of 4-(azepan-1-ylsulfonyl) -N- (pyridin-3-ylmethylideneamino) benzamide is O=C(NN=CC1=CC=CN=C1)C1=CC=C(S(=O) (=O)N2CCCCCC2)C=C1 ...</think>` |
| Answer | `<answer>O=C(NN=CC1=CC=CN=C1)C1=CC=C(S(=O) (=O)N2CCCCCC2)C=C1</answer>` ✓ |

The IUPAC-to-SMILES conversion task presented requires the model to translate a systematic chemical name into its corresponding SMILES string notation. We source our examples from the USPTO-480K dataset and prompt the model with a clear generative instruction "Given this IUPAC name, please provide the corresponding SMILES notation." The ground truth is established by the Tanimoto similarity score. The model's chain-of-thought reasoning is captured inside <think>...</think> tags and the final SMILES inside <answer>...</answer>. This setup tests the model's ability to parse complex nomenclature and generate SMILES structural representations.

For GRPO reward pipeline, we use accuracy reward function that only considers the final '<answer>' SMILES $s_{\text{ans}}$. After converting both the reference $s^*$ and prediction $s_{\text{ans}}$ to Morgan fingerprints, it computes the raw Tanimoto similarity $\tau$ and then applies a piecewise linear scaling:

$$r_{\text{tf}}(\tau) = \begin{cases} +1.0, & \tau = 1.0, \\ \tau - 0.3, & 0.3 \leq \tau < 1.0, \\ -0.5, & \tau < 0.3. \end{cases}$$

A perfect match yields $r_{\text{tf}} = 1.0$, moderate similarity is rewarded proportionally above $-0.5$, and very poor matches incur a negative penalty.

## B.3 REACTION NAMING

---

***Task: Reaction Naming (RxN)***

---

| | |
|---|---|
| Dataset Source | USPTO Reaction 1M; (Coley et al., 2017) |
| Input | What is the name of this chemical reaction? The reagents are `CC(C)(C)[Si](C)(C)Cl` and `Nc1ccc(-c2cccc(O)c2)c2c1C(=O)NC2` and the product `CC(C)(C)[Si](C)(C)Oc1cc cc(-c2ccc(N)c3c2CNC3=O)c1`. Choose ONLY from the following options and write your response choice inside `<answer>...</answer>`: "Acylation", "Aromatic Heterocycle Formation", "C-C Coupling", "Deprotection", "Functional Group Addition", "Functional Group Interconversion", "Heteroatom Alkylation and Arylation", "Miscellaneous", "Protection", "Reduction". Do not provide final answer different than what is provided in this list. |
| Target Molecular Reaction Name | Protection |
| Output | <think> First, let's break down the reagents. The first reagent is `CC(C)(C)[Si](C)(C)Cl` which is `tert-butyldimethylsilyl chloride` ... |
| Answer | <answer> Protection </answer> |

---

Table 5: Reaction Naming experiment description.

The reaction naming task is a classic example of a structured classification problem in cheminformatics, where the goal is to categories the nature of a reaction given reactants, conditions and products. This approach aim to test the ability of the LLM to conduct chemical reasoning and instruction following for discrete level answering. In addition to that, this setup also tests the model's ability to interpret chemical structures from linear notation and enables us to reveal how chain-of-thought guidance and prompt design impact classification accuracy. To stimulate reasoning, the model is tasked to output his thinking process inside <think>...</think> tags before emitting the final choice in <answer>...</answer> tags. The ground-truth class labels are evenly drawn from ten commonly found reaction type in chemistry: "Acylation", "Aromatic Heterocycle Formation", "C-C", "Coupling", "Deprotection", "Functional Group Addition", "Functional Group Interconversion", "Heteroatom". "Alkylation and Arylation", "Miscellaneous", "Protection" and "Reduction" derived from curated USPTO reactions dataset.

**Reward Functions:**

- **Continuous Format Reward:**
    - This reward is described in Section D.2.1 in the Algorithm 3.
- **Accuracy Reward:**
    - 0 if no answer is given
    - 0.1 if a single answer is given (but wrong)
    - 1 if the answer is entirely correct
    - -0.2 penalty if the model always output the same wrong class
- **Accuracy Percentage Reward:** discrete reward to foster perfect answers
    - 0 if the answer is wrong

– 1 if the answer is entirely correct

## B.4 Reaction Replacement

---

***Task: Reaction Replacement (RxR)***

---

| | |
|---|---|
| Dataset Source | USPTO Reaction 1M; (Coley et al., 2017) |
| Input | Question: Which chemical reaction is correct? Choose from the following options: |
| | A. In the following reaction, the reagents are: `Cc1ncc(C=O)n1C1CC1`, `CC(C)OC=C(Br)C=O`, `Cl`, `O=C(c1cc(N2CCNC2=O)ccc1F)N1CCCN(c2nccs2)CC1` and the product is: `O=Cc1cnc2n1CCCC2`. |
| | B. In the following reaction, the reagents are: `Cc1ncc(C=O)n1C1CC1`, `CC(C)OC=C(Br)C=O`, `Cl`, `N=C1CCCCN1` and the product is: `CNC(=O)CC1(O)CCCN(C(=O)c2cncc(F)c2)C1`. |
| | C. In the following reaction, the reagents are: `Cc1ncc(C=O)n1C1CC1`, `CC(C)OC=C(Br)C=O`, `Cl`, `N=C1CCCCN1` and the product is:`O=Cc1cnc2n1CCCC2`. |
| | D. In the following reaction, the reagents are:`Cc1ccccc1OCCC(=O)N1CCCC(c2ccn[nH]2)C1`, `CC(C)OC=C(Br)C=O`, `Cl`, `N=C1CCCCN1` and the product is: `O=Cc1cnc2n1CCCC2`. |
| | Make sure to give your choice A, B, C, or D inside the <answer>...</answer> tags. |
| Target Molecular Reaction (Choice) | C |
| Output | <think> Let's evaluate each option step by step to determine which one is correct. Option A: The reagent: `Cc1ncc(C=O)n1C1CC1` matches with the molecule `Cc1ncc(C=O)n1C1CC1`. |
| Answer | <answer> C </answer> |

Table 6: Reaction Replacement experiment description.

The reaction replacement tasks challenges the model to understand chemical reaction concepts, validity and ability to detect subtle structural inconsistencies. By providing the model with four nearly identical choices, chemical reaction notation coherence understanding is required. Each dummy reaction has one reagent randomly swapped, where starting from a correct USPTO reaction, we generate three "corrupted" variants by replacing a single reactant or product with the most Tanimoto-similar molecule drawn from a random batch of 50 Enamine50k compounds. In the prompt we provide the lists options A–D, each specifying reagent SMILES, conditions SMILES, and product SMILES, and the model is then instructed to answer one of the four choices as the correct one. The model is also instructed to think through each option step by step inside <think>...</think> and the answer is emitted inside <answer>...</answer> tags.

**Reward Functions:**

- **Continuous Format Reward:**
    - This reward is described in Section D.2.1 in the Algorithm 3.
- **Accuracy Reward:**
    - 0 if the answer is wrong
    - 1 if the answer is entirely correct

## B.5 Reaction Inversion

---

***Task: Reaction Inversion (RxI)***

---

| | |
|---|---|
| Dataset Source | USPTO Reaction 1M; (Coley et al., 2017) |
| Input | Question: Which chemical reaction is correct? Choose from the following options:

A. In the following reaction, the reagents are: `BrCc1ccccc1`, `[K+]`, `[OH-]`, `O=C(O)c1ccc(OCc2ccccc2)cc1` and the product is: `CCOC(=O)c1ccc(O)cc1`.

B. In the following reaction, the reagents are: `C=O`, `O=Cc1ccccc1`, `[B-]C#N`, `[Na+]`, `CN[C@H]1[C@@H](C)C[C@@H](c2ccncc2NC(=O)OC(C)(C)C)C[C@H]1NC(=O)OC(C)(C)C`, the conditions are: `CO`, `[OH-]`, `[OH-]`, `[Pd+2]`, and the product is: `C[C@H]1C[C@@H](c2ccncc2NC(=O)OC(C)(C)C)C[C@@H](NC(=O)OC(C)(C)C)[C@H]1N`.

C. In the following reaction, the reagents are: `CCOC(=O)C#N`, `CCOC(=O)Cl`, `Cc1ccoc1C=Nc1ccccc1`, the condition is: `C1(C)C(C)=CC=CC=1`, and the product is: `CCOC(=O)c1cc2ccoc2cn1`.

D. In the following reaction, the reagents are: `CC1(C)OB(c2cn[nH]c2)OC1(C)C`, `Nc1nc(-c2cc3c(s2)-c2ccc(-c4cn[nH]c4)cc2OCC3)c(-c2ccccc2Cl)s1` and the product is: `Nc1nc(-c2cc3c(s2)-c2ccc(Br)cc2OCC3)c(-c2ccccc2Cl)s1`.

Make sure to give your choice A, B, C, or D inside the <answer>...</answer> tags. |
| Target Molecular Reaction (Choice) | C |
| - - - - - - - - - - - - - - - - - - | - - - - - - - - - - - - - - - - - - - - - - - - - - - - |
| Output | <think> Starting with option A: The reaction uses benzyl bromide `BrCc1ccccc1` ... |
| Answer | <answer> C </answer> |

Table 7: Reaction Inversion experiment description

The reaction inversion task challenges the model to understand chemical reaction concepts, validity and ability to detect subtle structural inconsistencies. By providing the model with four completely different choices, strong chemical reaction notation coherence understanding is required. Each dummy reaction has one reagent randomly swapped with the longest string SMILES among the products, enabling us to obtain 4 different reaction choices. In the prompt we provide the lists options A–D, each specifying reagent SMILES, conditions SMILES, and product SMILES, and the model is then instructed to answer one of the four choices as the correct one. The model is also instructed to think through each option step by step inside <think>...</think> and the answer is emitted inside <answer>...</answer> tags.

**Reward Functions:**

- **Continuous Format Reward:**
  - This reward is described in Section D.2.1 in the Algorithm 3.
- **Accuracy Reward:**
  - 0 if the answer is wrong
  - 1 if the answer is entirely correct

## B.6 REACTION TRUE/FALSE

---

***Task: Reaction True/False (RxTF)***

---

| | |
|---|---|
| Dataset Source | USPTO Reaction 1M; (Coley et al., 2017) |
| Input | Question: Is this chemical reaction correct? In the following reaction, the reagent is: `COC(=O)c1ccc(OC)c(OCCc2cccc(C#N)c2)c1`, the conditions are: `C1COCCO1, [Li+], [OH-]`, and the product is: `COc1ccc(C(=O)O)cc1OCCc1cccc(C#N)c1`. |
| Target Molecular Reaction Validity | True |
| Output | \<think\> First, I remember that LiOH, `[Li+] . [OH-]` is a strong base, so it's likely an acid-base reaction. The ester group in the starting material ... |
| Answer | \<answer\> True \</answer\> |

Table 8: Reaction True/False experiment description

The Reaction True/False task is a binary derivative of the Reaction Replacement task. In this case, the model is asked to analyze and judge based on one single reaction, whether the reaction is correct or wrong. Each prompt presents one reaction—listing the reagent SMILES, the reaction conditions SMILES, and the product SMILES—and then asks "Is this chemical reaction correct?". The examples are drawn from the Reaction Replacement set, where some of the reactions have been corrupted by swapping one random molecule in the reaction string by a new candidate. The model is instructed to reason step by step inside \<think\>...\</think\>, then has to emit \<answer\>True\</answer\> or \<answer\>False\</answer\> accordingly. This format was designed to simplify the reaction replacement task by providing only a binary label choice, allowing us to not only reduce the task complexity but also diminish the hallucination effects emanating from providing many examples in the prompt. In this scenario, the model only ever sees one reaction, and its ability to detect subtle mismatches in chemical transformations and to follow a true/false classification protocol with transparent chain-of-thought is tested.

**Reward Functions:**

- **Continuous Format Reward:**
    - This reward is described in Section D.2.1 in the Algorithm 3.
- **Accuracy Reward:**
    - 0 if the answer is wrong
    - 1 if the answer is entirely correct

## B.7 CHEMICAL FORMULA BALANCING TASK (CEB)

***Task: Chemical Formula Balancing Task (CeB)***

| | |
|---|---|
| Dataset Source | The Perovskite Dataset (Jacobsson et al., 2022) |
| Input | You are now acting as a skilled chemistry expert. Your goal is to identify and complete the missing compound(s) represented by [MASK] in the given chemical reaction. Reaction: 6 BaCO3 + 1 Na2CO3 + 4 SiO2 == [Mark] Na2Ba[Mark](Si[Mark]O[Mark])(SiO[Mark])2 + 7 CO2. Analyze the equation carefully, considering the law of conservation of mass and stoichiometry. Ensure the total number of atoms of each element on both sides of the equation remains balanced. |
| Target Molecule (Material) | 6 BaCO3 + 1 Na2CO3 + 4 SiO2 = 1 Na2Ba6(Si2O9)(SiO3)2 + 7 CO2 |
| Output | <thinking>1. Element Count Analysis: - Reactants: - Ba: 6 (from 6 BaCO3) - Na: 2 (from Na2CO3) - Si: 4 (from 4 SiO2) - C: 7 (6 from BaCO3 + 1 from Na2CO3) - O: 6×3 (BaCO3) + 1×3 (Na2CO3) + 4×2 (SiO2) = 29 - Products: - CO2: 7 CO2 (7 C, 14 O) - Missing compound must account for remaining elements: 6 Ba, 2 Na, 4 Si, and 15 O (29 total O - 14 from CO2). 2. Structure of Missing Compound: - The formula template [MASK] Na2Ba[Mark](Si[Mark]O[Mark])(SiO[Mark])2 suggests: - 2 Na, 6 Ba (to balance Ba from reactants), 4 Si (from SiO2), and 15 O. - Silicate groups: (Si2O9) and (SiO3)2 (to total 4 Si and 15 O). 3. Charge Balance: - Cations: Na2 (2+) + Ba6 (12+) = +14 charge. - Anion: Si4O15 (4 Si × +4 + 15 O × -2 = +16 -30 = -14) balances the +14 charge. 4. Final Formula: - Na2Ba6(Si2O9)(SiO3)2 consolidates all elements and charges correctly.</thinking> |
| Answer | <answer>6 BaCO3 + 1 Na2CO3 + 4 SiO2 = 1 Na2Ba6(Si2O9)(SiO3)2 + 7 CO2</answer> |

Table 9: Template for reporting RL experiments.

The goal of this task is to evaluate the performance of large language models (LLM) in predicting balanced chemical equations, specifically leveraging a reward mechanism to enhance the accuracy of chemical formula generation. The GRPO reinforcement learning data set was derived from a data set of chemical formulas focusing on perovskite compounds. The dataset was prepared by selectively masking certain products or their stoichiometric coefficients within balanced chemical equations.

**Reward Function Design:** The reward function combines two distinct components: format correctness and prediction accuracy.

- **Format Reward:**
    - Initial reward is set to 0 if the generated chemical equation does not follow the correct format.
    - An additional reward of 0.3 is given if the left-hand side (reactants) of the generated equation matches exactly with the ground truth.
- **Accuracy Reward:** This reward component assesses the similarity between the generated right-hand side (products) ($y_{LLM}$) and the ground truth ($y_{GT}$) using a predefined similarity metric, formulated as:

$$\text{reward} = \text{format\_reward} + \text{sim}(y_{LLM}, y_{GT})$$

## B.8 CONDITIONAL MATERIAL GENERATION (CMG)

| *Task: Conditional Material Generation (CMG)* | |
| --- | --- |
| Dataset Source | Materials Project (Jain et al., 2013) |
| Input | Build a material that has O, Te, Tm. Please keep your reasoning as concise as possible. For example <material> A A B B B <sg12></material> where A, B refer to elements and <sg12> denotes the space group for example: <material> Pa In Tc Tc <sg225></material>. |
| Target Material Composition | Any valid composition made up of the elements O, Te and Tm with a valid space group number e.g. <material> O O Te Tm Tm Te <sg127></material> |
| Output | To combine O, Te, and Tm, a space group like sg127 (for a 1:1:1 ratio) could work. A possible structure could be: <material> O O Te Tm Tm Te <sg127></material> This fits the space group 127, which is a common space group for spacefilling structures.(Note: Space groups can vary, but this is a reasonable choice for a simple ternary compound.) |
| Answer | <material> O O Te Tm Tm Te <sg127></material> |

Table 10: Conditional Material Generation experiment description

This task aims to leverage the scientific knowledge embedded in MiST-trained LLMs to generate novel materials from a specified set of elements. The experiment focuses on the model's ability to understand three-dimensional atomic relationships within crystal structures and, based on that understanding, produce valid compositions. If the model can perform this task with high accuracy, it could significantly enhance the efficiency and cost-effectiveness of the material generation phase in the materials discovery process.

**Reward Function Design:** The quality of the generated composition is measured by the metrics: validity, precision and novelty. Validity is assessed using SMACT (Davies et al., 2016) validity, which checks whether the generated composition adheres to fundamental chemical rules, such as charge neutrality. Precision measures the model's ability to follow instructions and correctly include the specified elements. It is computed using the following equation:

$$\text{Precision} = \frac{|E_{pi} \cap E_{qi}|}{E_{pi}},$$

where $E_{pi}$ is the set of elements specified in the $i$-th prompt and $E_{qi}$ is the corresponding generated element (Xia et al., 2025). The novelty of the generated composition was determined based on whether the composition was present within the materials project dataset or was previously generated by the model. Furthermore, to ensure the model provided its generated solution in a valid format, the reward function also checked that the generated composition was enclosed within the <material>...</material> tags and that the assigned space group number lies within the valid range of 1 to 230.

Therefore, the reward function used to train the LLM for the conditional material generation task was:

$$R = \alpha_1 \text{Validity} + \alpha_2 \text{Precision} + \alpha_3 \text{Novelty} + \alpha_4 \text{Format},$$

where the parameters $\alpha_i$ for $i = 1, ..., 5$ are scaling factors for each portion of the reward.

## B.9   BINARY COMPOUND STRUCTURE RELAXATION TASK (CRR)

---

***Task: Binary Compound Structure Relaxation Task (CrR)***

---

| | |
|---|---|
| Dataset Source | Materials Project Trajectory (MPtrj) Dataset (Deng, 2023), Materials Project (Jain et al., 2013) |
| Input | Given a perturbed binary compound: formula Ti 1_int Ni 1_int, space_group_symbol 'P1' lattice_parameters a 2.74603000 b 2.94419691 c 2.94419691 alpha 119.99997545 beta 90.00000000 gamma 90.00000000 Zr 1_int 0.00000000 0.00000000 0.00000000 N 1_int 0.50000000 0.66670000 0.33330000 perform multiple steps Structural Relaxation on the given perturbed binary compound and reduce the internal energy. Please document your thought process within <think> </think> tags, and provide the final corrected structure in <answer> </answer> tags |
| Target Material | A more stable structure of the same material with lower inner energy. |
| Output | <thinking>The input structure describes ZrN in a hexagonal P-6m2 space group with significant initial distortions. The relaxation process improves stability through three key mechanisms: 1. Symmetry Restoration: The nitrogen position (0.5,0.6667,0.3333) deviated from standard WC-type coordinates (1/3,2/3,1/2). Relaxation shifts N to (0.3333,0.6666,0.5), aligning with proper Wyckoff positions for P-6m2. This restores three-fold rotational symmetry about [001], eliminating unnecessary symmetry-breaking distortions.2. **Bond Strain Reduction**: Initial Zr-N bonds (2.19 Å) were compressed relative to ionic radius predictions ($Zr^{3+}$ CN6: 0.85 Å+ $N^{3-}$ CN6: 1.32 Å= 2.17 Åideal). However, the metallic character of ZrN allows for shorter bonds. The relaxation increases bond length to 2.34 Å, balancing ionic size mismatch with metallic bonding contraction while reducing internal strain energy.3. **Lattice Parameter Optimization**: The original a-axis (2.746 Å) expanded to 3.15 Åpost-relaxation, better matching hexagonal close-packing requirements for Zr (atomic radius 1.60 Å). The c-axis remained near 2.94 Å, maintaining axial ratio c/a ≈ 0.93 - consistent with known WC-type structures rather than ideal HCP (c/a=1.633).The final structure shows improved prototype alignment with tungsten carbide (space group P-6m2), where both elements occupy high-symmetry positions. However, residual distortion in the pentagonal pyramidal coordination suggests possible metastability compared to the more common rock-salt ZrN polymorph. The increased symmetry and reduced bond strain would lower total energy, though the hexagonal phase may still lie above the cubic phase in energy.</thinking> |
| Answer | formula Zr 1_int N 1_int space_group_symbol 'P1' lattice_parameters a 3.15089295 b 3.15093266 c 2.93526192 alpha 90.00000000 beta 90.00000000 gamma 119.99958321 Zr 1_int 0.99999066 0.99998132 0.00000000 N 1_int 0.33330934 0.66661868 0.50000000</answer> |

Table 11: Template for reporting RL experiments.

---

This task aims to utilize LLMs to perform preliminary relaxation (pre-relaxation) of unstable crystal structures, aiming to yield more stable structural configurations. This preliminary step is intended to substantially decrease computational costs and improve efficiency in subsequent high-accuracy Density Functional Theory (DFT) calculations. DFT calculations, while accurate, are computationally intensive. By leveraging LLM-generated pre-relaxation adjustments, the experiment seeks to effectively reduce the quantity of computationally unfavorable structures, thereby streamlining and accelerating the DFT computational pipeline.

**Format Reward:**

$$R_{\text{format}} = \begin{cases} -1, & \text{if } S_{\text{gen}} \text{ is valid Mat2Seq format and have lower inner energy than input structure} \\ -5, & \text{if } S_{\text{gen}} \text{ is valid Mat2Seq format} \\ -10, & \text{otherwise} \end{cases}$$

## C  BENCHMARKING PROCEDURE

In this section we elaborate on the methods used to evaluate the models in the multiple ways displayed in Table 20. Here we give details of how diagnostic metrics have been computed (SCS, CCS), which evaluate the capabilities in LLMs that are necessary for success on chemical tasks in an RL setting. Additionally, performances on downstream tasks have been computed using benchmarks derived from each task (see Appendix above), along with different prompting techniques, that mark the difference between direct answer, or reasoning answer.

### C.1  LATENT SYMBOLIC AND CHEMICAL KNOWLEDGE

#### C.1.1  SYMBOLIC COMPETENCE SCORE BENCHMARK

The Symbolic Competence Score benchmark measures the model's latent capability to read and write correct chemical symbols. In this benchmark we focus particularly on SMILES, as organic chemistry spans a majority of our tasks. For this we collected 10000 valid SMILES from PubChem Kim et al. (2025), such that no overlap exists with the MiST data. A second dataset is created with corrupted smiles based on these smiles, where corruptions are minimal, however render the smiles unvalid. The corruption procedure is specified in Algorithm 1. The algorithm removes a random subset of key structural grammar elements (ring/branch brackets and digits) from the SMILES string, producing broken or ambiguous strings. Corruption rate $\rho$ controls the proportion of removed elements, which for all our experiments has been set to 0.2.

---

**Algorithm 1:** SMILES Grammar Element Corruption

---

**Input:** SMILES string $s$, corruption rate $\rho$
**Output:** Corrupted SMILES string $s_{\text{corrupt}}$
   Let $\mathcal{G} = \{\,(\,,\,)\,,\,[\,,\,]\,, 0, 1, 2, 3, 4, 5, 6, 7, 8, 9\}$ (grammar elements);
$L \leftarrow$ length of $s$;
$I \leftarrow$ indices of $s$ where $s_i \in \mathcal{G}$;
**if** $|I| = 0$ **then**
   |   **return** $s$;
**end**
$N_{\text{remove}} \leftarrow \max(1, \lfloor \rho \cdot |I| \rfloor)$;
Randomly select $R \subseteq I$ with $|R| = N_{\text{remove}}$;
$s_{\text{corrupt}} \leftarrow$ empty string;
**for** $i \leftarrow 1$ **to** $L$ **do**
   |   **if** $i \notin R$ **then**
   |     |   Append $s_i$ to $s_{\text{corrupt}}$;
   |   **end**
**end**
**return** $s_{\text{corrupt}}$;

---

Finally, evaluation happens in two stages. First, the log-likelihoods are computed using the model for the following string, that provides context for the string to look more natural:

```
The molecule represented with the SMILES
[BEGIN_SMILES] smiles [END_SMILES]
```

Where `smiles` is replaced by both the correct, and the incorrect SMILES string. The log-likelihoods corresponding to the smiles tokens are isolated by dropping the computed likelihoods associated with the context shown above. The two corresponding strings are thus

Original SMILES:

```
The molecule represented with the SMILES
[BEGIN_SMILES] O=C(O)C[C@H](O)C[C@H](O)CCn2c(c(c(c2c1ccc(F)cc1)c3ccccc3)C(=O)N
[END_SMILES]
```

Corrupted SMILES:

```
The molecule represented with the SMILES
[BEGIN_SMILES] O=C(O)C[C@H](O)C[C@H](O)CCn2c(c(c(c2c1ccc(F)cc1)c3ccccc3)C(=O)N
[END_SMILES]
```

Average loglikelihoods are computed for the whole sample of 10000 SMILES in this manner, and SCS score is computed as the Cohen's d effect size between the distributions of loglikelihoods of correct smiles, vs that of corrupted smiles.

Note that although the structure of material compositions is different from that of SMILES, the corruption method is similar, as key structural elements such as the space group number tag (`<sg12>`) and elemental symbols are replaced with special characters.

### C.1.2 Chemical Competence Score benchmark

The Chemical Competence Score (CCS) evaluates a model's latent ability to distinguish between chemically accurate and inaccurate factual statements. To construct this benchmark, we selected 1,000 samples from the test split of the SMolInstruct Molecule Description dataset (Yu et al., 2024b), which was never used in all post-training stages. Each sample in the dataset consists of a brief description of an organic molecule. For example, one entry describes an acetamide as:

```
N-[4-(1,3-thiazol-2-ylsulfamoyl)phenyl]acetamide is a
sulfonamide that is benzenesulfonamide substituted
by an acetylamino group at position 4 and a
1,3-thiazol-2-yl group at the nitrogen atom.  It is
a metabolite of sulfathiazole.  It has a role as a
marine xenobiotic metabolite.  It is a sulfonamide, a
member of acetamides, and a member of 1,3-thiazoles.
```

For material data, we utilized `Robocrystallographer` (Ganose & Jain, 2019) to generate 600 natural text descriptions for crystal structures from the Material Project (Jain et al., 2013). Here is an example entry:

```
AlN is Wurtzite structured and crystallizes in the
hexagonal P6_3mc space group.  Al(1) is bonded to
four equivalent N(1) atoms to form corner-sharing
AlN4 tetrahedra.  There are three shorter (1.90 Å)
and one longer (1.91 Å) Al(1)-N(1) bond length.  N(1)
is bonded to four equivalent Al(1) atoms to form
corner-sharing NAl4 tetrahedra.
```

To create a contrastive benchmark, we generated an incorrect version for each entry by replacing one sentence in the original description with a sentence from a different one, while keeping the target molecule/crystal unchanged. Here is an example of an incorrect version of the above acetamide example with the edited section highlighted:

```
N-[4-(1,3-thiazol-2-ylsulfamoyl)phenyl]acetamide is a
tricyclic triterpenoid of the isomalabaricane group.
It is a metabolite of sulfathiazole.  It has a role as
a marine xenobiotic metabolite.  It is a sulfonamide,
a member of acetamides and a member of 1,3-thiazoles.
```

### C.2 Task Benchmarks

The benchmarks have been obtained by selecting a subset of the datasets defined in Appendix B, for each of the tasks.

### C.3 Inference techniques

We observed that models' full text generation often overflows the available context window, without providing any final answer within <answer> tags, thus preventing its correct evaluation. To overcome

Table 12: Evaluation methods for each reaction task

| Task | Evaluation Method |
|---|---|
| Reaction Prediction (RxP) | Exact match with the groundtruth product |
| Reaction Naming (RxN) | Top-1 classification accuracy over the 10 reaction classes. |
| Reaction Replacement (RxR) | Multiple-choice accuracy (selecting the one correct reaction out of four). |
| Reaction Inversion (RxI) | Multiple-choice accuracy (selecting the one correct reaction out of four). |
| Reaction True/False (RxTF) | Binary classification accuracy (correct vs. incorrect reaction). |

this, upon failure to generate an <answer> tag, we directly append the <answer> tag and retry the generation, biasing the model towards generating an answer at that point. Pseudo-code for this is provided in Algorithm 2.

An extension of such an injection technique is that models can be biased from the beginning of the completion towards directly providing an answer, thereby allowing us to evaluate the effect of the intermediate text inside <think> tags. In Table 20 in the main manuscript, direct answer results are reported outside of the parentheses, while reasoning results are in parentheses.

---

**Algorithm 2:** Answer tag injection `<answer>` - Think and answer procedure

---

**Input** : prompt, $model\_sampling\_params$, model, $nbr\_max\_retries$
**Output** : A completion containing `<answer>...</answer>`

result $\leftarrow$ llm.generate(prompt, sampling_params);
completion $\leftarrow$ result.outputs[0].text;
**for** $i \leftarrow 1$ **to** $max\_retries$ **do**
    `// Append the '<answer>' token to coax a proper tag`
    new_prompt $\leftarrow$ prompt ++ competition ++ `"<answer>"`;
    result $\leftarrow$ llm.generate(new_prompt, sampling_params);
    complete_completion $\leftarrow$ result.outputs[0].text;
    **if** *HasAnswer(complete_completion )* **then**
        **return** complete_completion;

**return** complete_completion ;        `// fallback if still no tag`

---

## D EXPERIMENTAL SETTINGS

### D.1 MIST: MID-STAGE SCIENTIFIC TRAINING

Our MiST model is based on the Qwen-2.5-3B model. We continue the pre-training and perform SFT thereafter on a chemically enriched corpus spanning a diversity of sources, targeting the two prerequisites we proposed in the main manuscript.

The following configuration of hyperparameters was used for training:

Table 13: MiST Pretraining Hyperparameters

| Parameter | Value |
|---|---|
| Model Architecture | Qwen-2.5-3B |
| Epochs | 4 ($\sim$90,000 steps) |
| Batch Size | 32 |
| Max/Min Learning Rate | $1 \times 10^{-5}$ / $1 \times 10^{-6}$ |
| LR Warmup Steps | 1,000 |
| LR Decay Steps | 1,000 |
| Optimizer | AdamW |
| Loss Function | Cross-Entropy |
| Hardware | $32 \times$ H100 GPUs |
| Total GPU Hours | 640 |

After this stage, the model is further trained with SFT on instruction and Q&A data, as well as reasoning traces obtained from a stronger reasoning LLM, on more chemistry-relevant tasks; see the following section for more details. The following configuration was used:

Table 14: MiST SFT Hyperparameters

| Parameter | Value |
|---|---|
| Model Architecture | Qwen-3B |
| Epochs | 3 ($\sim$32,000 steps) |
| Batch Size | 32 |
| Learning Rate | $1 \times 10^{-6}$ |
| Optimizer | AdamW |
| Loss Function | Cross-Entropy |
| Hardware | $32 \times$ H100 GPUs |

## D.2 REINFORCEMENT LEARNING EXPERIMENTS

The Open-R1 repository from Hugging Face (`https://github.com/huggingface/open-r1`) was forked and modified with additional features/optimizations for the GRPO experiments. Each training was run for 12 hours on four nodes (with four NVIDIA GH200 120GB GPUs), summing to 16 GPUs and 192 GPU-hours per training. The best hyperparameters are summarized in Table 15. A completion length of 8192 was used to let the model output long reasoning thoughts. The best hyperparameters and rewards were optimized using a total of 30k GPU-hours with variations in the experimental setups. The list of used rewards is described in Section D.2.1.

| parameter | value |
|---|---|
| per_device_train_batch_size | 1 |
| gradient_accumulation_steps | 8 |
| learning_rate | 2e-6 |
| lr_scheduler_type | cosine |
| warmup_ratio | 0.03 |
| beta | 0.04 |
| max_prompt_length | 384 |
| max_completion_length | 8192 |
| num_generations | 8 |
| use_vllm | true |
| vllm_max_model_len | 8192 |

Table 15: Optimized hyperparameters used for the GRPO training experiments.

### D.2.1 REWARDS

The rewards designed for our GRPO experiments are grouped into two main categories:

- Format reward: the goal is to ensure that the trained model uses the appropriate format with reasoning (between <think> tags) and answer (between <answer> tags).

- Accuracy reward: the goal is to verify the answer of the model for the given task.

**Accuracy reward**: For the different tasks, different accuracy rewards are implemented in a continuous manner if possible. For SMILES-based tasks, the Tanimoto similarity score is generally used. However, for MCQA-based tasks, the rewards are usually discrete since the answers are correct or wrong. These rewards typically range from 0 to 1 (perfect answer).

**Accuracy percentage reward**: For each task, we also implement a discrete accuracy percentage reward to foster perfect answers and to log the training accuracy of the models. This reward is 0 if the answer is wrong and 1 if the answer is entirely correct.

**Continuous format reward**: A continuous format reward has been implemented with the structure described in Algorithm 3. The idea behind this reward is to output a score between -1 (very bad format)

and 1 (perfect format) with continuous steps to help the model with the learning of the expected format.

---

**Algorithm 3:** Incremental Formatting Reward Calculation

---

**Input**   :Raw model output $o \in$ `String`
**Output**:Formatting reward $r \in [-1, 1]$

$r \leftarrow 0.0$                                                                                    `// Initialize reward`
$T \leftarrow \{$`<think>`,`</think>`, `<answer>`, `</answer>`$\}$
`// Check each tag appears exactly once`
**foreach** tag $\in T$ **do**
  **if** $\text{COUNT}(o, \text{tag}) = 1$ **then** $r \leftarrow r + 0.05$
  **else** $r \leftarrow r - 0.05$
**end**

`// Check correct start and end tags`
**if** $\text{STARTS\_WITH}(o, $`<think>`$)$ **then** $r \leftarrow r + 0.05$
**else** $r \leftarrow r - 0.05$
**if** $\text{ENDS\_WITH}(o, $`</answer>`$)$ **then** $r \leftarrow r + 0.05$
**else** $r \leftarrow r - 0.05$

`// Check think-answer boundary`
**if** $\text{COUNT}(o, $`</think>`$\backslash n$`<answer>`$) = 1$ **then** $r \leftarrow r + 0.1$
**else** $r \leftarrow r - 0.1$

`// Check answer block extraction`
$m_1 \leftarrow \text{REGEX\_MATCH}($`<answer>`$(.*)$`</answer>`$, o)$
**if** $m_1 = \text{None}$ **then**
  | $r \leftarrow r - 0.2$
**else if** $\text{NUM\_GROUPS}(m_1) \neq 1$ **then**
  | $r \leftarrow r - 0.05$
**else**
  | $r \leftarrow r + 0.2$
**end**

`// Check whole think \n answer pattern`
$m_2 \leftarrow \text{REGEX\_MATCH}($`<think>`$(.*)$`</think>`$\backslash n$`<answer>`$(.*)$`</answer>`$, o)$
**if** $m_2 = \text{None}$ **then**
  | $r \leftarrow r - 0.4$
**else if** $\text{NUM\_GROUPS}(m_2) \neq 2$ **then**
  | $r \leftarrow r - 0.1$
**else**
  | $r \leftarrow r + 0.4$
**end**

**return** $r$

---

# E DATA

## E.1 DATA SOURCES AND PROCESSING

### E.1.1 FINEWEB-EDU

The FineWeb-Edu can be found on Hugging Face (`https://huggingface.co/datasets/HuggingFaceFW/fineweb-edu`) (Penedo et al., 2024). The subsets "CC-MAIN-2013-20" to "CC-MAIN-2024-10" were downloaded for a total of ∼6 TB, which represents roughly 1.3T tokens and 1.26B individual texts. Based on the representative subset "sample-10BT" (also downloaded), the text sources were computed by taking the base URL (from the dataset column "url"), then these sources were sorted from the most prevalent to the least. We manually labeled the most prevalent sources as "chemistry", "non-chemistry", or "undetermined". The goal was to label a source as "chemistry" only if nearly all the texts from that source are about chemistry. On the other hand, a source is classified as "non-chemistry" only if there is no mention of chemistry in all the texts from that source. When a source contains a mix, like a school website with chemistry texts and texts for other fields, the label used is "undetermined", and the source is not used. After this manual labeling, the texts from "sample-10BT" were classified based on the labeled sources. It led to a ground truth of approximately 10,000 "chemistry" texts and 50,000 "non-chemistry" texts (out of the ∼10M texts found in "sample-10BT"). Based on this ground truth, a custom non-ML classifier was built using the word frequencies in "chemistry" and "non-chemistry" texts. The texts were lemmatized before building word frequency vectors for the two classes using a simple processing script that replaces any non-standard character with a space, before splitting the strings by the spaces. A custom vocabulary was also built to store these lemmatized texts in a tokenized manner. Other lemmatization methods (such as Spacy or NLTK) were also tried, but did not lead to better results and were extremely expensive to use on the full FineWeb dataset (>6 TB). After building the vocabulary and the word frequency vectors for the two classes, the formula below was applied to each FineWeb text to create an associated "chemistry score" (ranging from 0 to "infinity"). The frequencies of the lemma $k$ in chemistry texts and non-chemistry texts are denoted $f_k^c$ and $f_k^n$, respectively. The text chemistry score (TCS) is computed using the following equation:

$$TCS(text) := \frac{1}{N_{lemmas}} \sum_{\substack{k=\text{lemma} \\ \text{in text}}} w_k \quad \text{with} \quad w_k = \begin{cases} f_k^c/f_k^n, & \text{if } f_k^c/f_k^n > 1 \\ 0, & \text{otherwise} \end{cases} \tag{4}$$

This labeling strategy was applied to the entire FineWeb-Edu corpus, and the texts with $TCS > 4$ were retained, yielding a pretraining set of 1.4 billion tokens of high-quality chemistry-labeled texts. The threshold $TCS > 4$ was decided based on the PR curve plot shown in Figure 7. This threshold allows for high precision, and the quantity of texts retrieved was sufficient for our pretraining pipeline. Additional plots with the percentage of chemistry texts by threshold and the cumulative number of chemistry token counts by threshold can be observed in Figures 8 and 9, respectively. Some chemistry text examples (with their associated TCS scores) are shown in Figure 10.

### E.1.2 ACADEMIC PAPER EXTRACTION

An overview of our preprocessing pipeline is depicted as follows. Initially, we leveraged Nougat (Blecher et al., 2023) and GROBID (Meuschke et al., 2023) libraries for converting PDF documents into textual formats. Nougat demonstrated superior performance in accurately transforming complex structures such as tables, formulae, bibliographic references, and figure captions into LaTeX-formatted text. Conversely, GROBID excelled at extracting plain textual content from PDFs. The output of the authors were merged with explicit tags assigned to each structural element: tables were encapsulated with [START_TABLE] and [END_TABLE], formulas marked by [START_FORMULA] and [END_FORMULA], bibliographic references enclosed within [START_BIBREF] and [END_BIBREF], and figure descriptions bracketed by [START_FIGURE] and [END_FIGURE]. Subsequently, this structured text was processed through the Chemical Data Extractor 2 (Swain & Cole, 2016), identifying candidate molecule entities along with their positional context within the text. To ensure high precision in entity identification, candidates were further validated using a custom-trained sentence transformer model, designed specifically to discern genuine molecular entities from contextual information. Validated molecular entities were then translated from their IUPAC nomenclature to SMILES notation using py2opsin, a Python interface for OPSIN

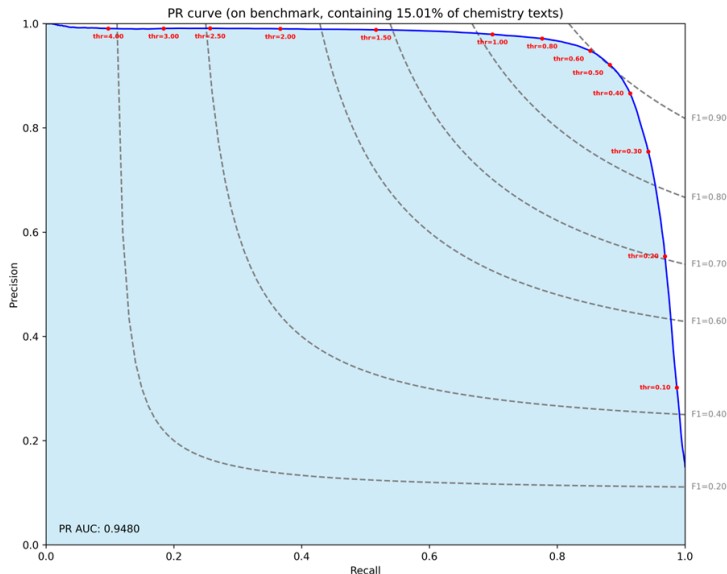

Figure 7: Precision-recall curve of the estimated retrieved chemistry texts based on the manually labeled ground truth. The different $TCS$ thresholds are shown in red dots on the PR curve.

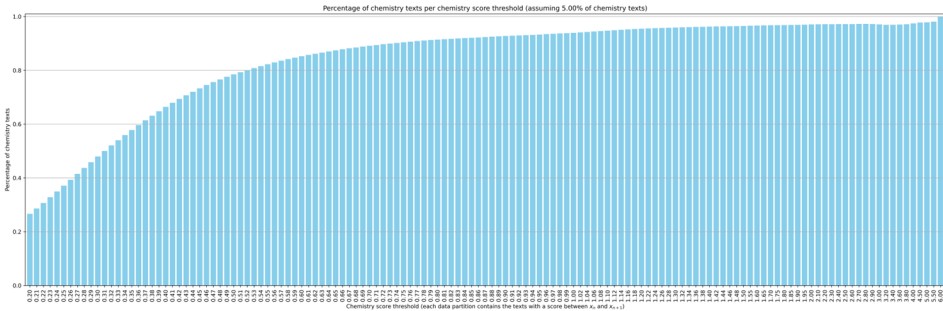

Figure 8: Estimated percentage of chemistry texts by TCS threshold.

(Lowe et al., 2011). In cases where OPSIN failed to yield a definitive conversion, entities were cross-referenced against PubChem (Kim et al., 2025). Ultimately, during the pretraining phase alone, our model encountered approximately 800,000 unique chemical compounds along with their corresponding SMILES representations.

### E.1.3 PUBCHEM

The first three million compounds from the PubChem database Kim et al. (2025) (CID from 1 to 3,000,000) were dumped using the PUG REST API with batched requests in October 2024. Each record contains these columns (among others): CanonicalSMILES, IsomericSMILES, IUPACName, and InChI. Since the molecule canonicalization algorithm used in the PubChem database is not the same as the one used by RDKit, all the compounds were re-canonicalized. The canonical SMILES consistency was also ensured for each compound by computing four canonical SMILES for each molecule:

- CanonicalSMILES → canonicalized using RDKit.
- IsomericSMILES → canonicalized using RDKit.
- IUPACName → SMILES using py2opsin and then canonicalized using RDKit.
- InChI → canonical SMILES using RDKit.

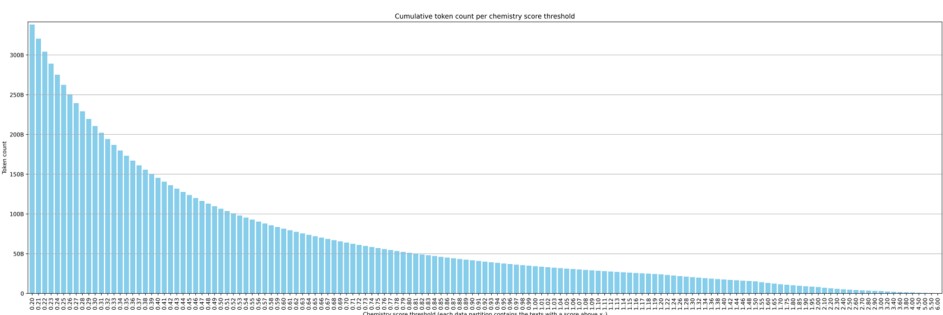

Figure 9: Estimated cumulative chemistry token count by TCS threshold.

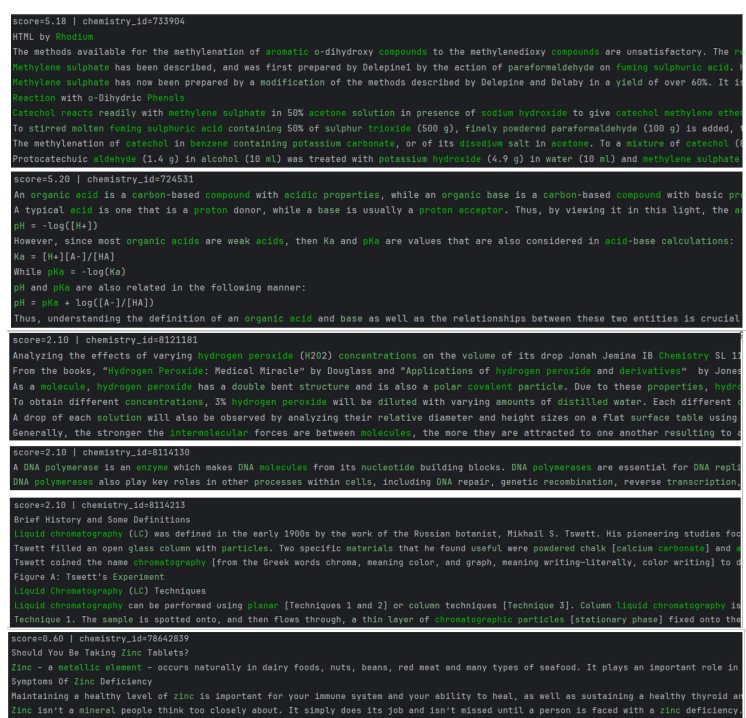

Figure 10: Examples of labeled chemistry texts with the associated TCS scores.

Then the four newly generated canonical SMILES were compared, and if a mismatch is found, the compound is discarded. This method filtered out approximately 40% of the compounds, and the duplicated canonical SMILES were also discarded. For the remaining compounds, four "SMILES variants" were computed using RDKit based on the canonical SMILES to have four non-canonical

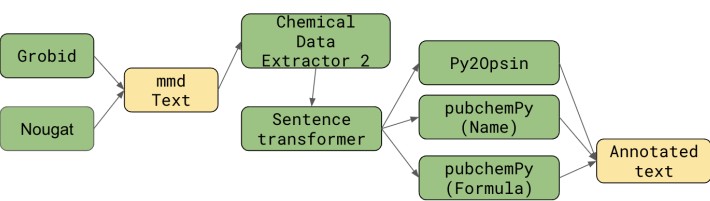

Figure 11: Overview of the preprocessing pipeline

SMILES in each record. At the end of this processing script, an approximate of 1,800,000 compounds were kept and ready to be used. The dataset was then split in the following manner: the first million compounds (CID from 1 to 1,000,000) were used for pretraining, the second million compounds (CID from 1,000,001 to 2,000,000) were used for GRPO training, and the third million compounds (CID from 2,000,001 to 3,000,000) were used as the test split for benchmarking. Each split contains ~600,000 valid compounds. Multiple derived datasets were also generated for the different chemical tasks used with GRPO training (explained in Section E.2 below).

## E.2 Chemical Tasks Data sources

All MCQA-derived tasks for GRPO training are built on the USPTO Reaction 1M dataset, and the I2S dataset was built using the PubChem dataset from Section E.1.3:

**Reaction Prediction (RxP)**

- The USPTO-480K dataset (Coley et al., 2019) consists of approximately 480K organic reactions, divided into training and test splits.
- We retained only reactions with a single product, resulting in roughly 400K training samples and 38K test samples.
- The first 10K reactions from the training set are used to generate reasoning traces.
- An additional 50K reactions, randomly selected from the remaining training data, are used for RLVF.
- A set of 500 reactions, randomly sampled from the test set, is used for benchmarking.

**IUPAC to SMILES (I2S)**

- The processed PubChem compounds (CID from 1,000,001 to 2,000,000) from the Section E.1.3 are used as the base data.
- The canonical SMILES and the IUPAC were directly used from the dataset.

**Reaction Naming (RxN)**

- Start from the full USPTO 1M reaction set.
- Use Rxn-Insight's class generation to detect the reaction name.
- Filter to 600 000 samples, evenly distributing across the 10 classes.

**Reaction Replacement (RxR)**

- Duplicate each USPTO 1M reaction four times.
- For three copies, randomly select one molecule (reactant or reagent) to replace.
- Draw a batch of 50 candidate molecules from Enamine50k and compute Tanimoto similarity.
- Swap in the most similar molecule as the replacement.

**Reaction Inversion (RxI)**

- Take four instances of reactions in USPTO 1M, and invert one reagent with a product for 3 of them.
- The LLM is required to predict which one of the four reactions is still correct.

**Reaction True/False (RxTF)**

- Derived from the Reaction Replacement dataset.
- Present a single reaction (original or corrupted) and ask the model to judge its chemical correctness.

## E.3 Material Tasks Data sources

**Chemical Formula Balancing Task (CeB)**

- A total of 1500 chemical formulas were selected from the Perovskite Dataset (Jacobsson et al., 2022) to form the data set, and the data set was then enhanced by randomly masking individual stoichiometric coefficients within products or entire product compounds using [MASK].

**Conditional Material Generation (CMG)**

- We selected 1000 samples from Materials Project (Jain et al., 2013) and extracted the constituent elements from each sample to create our dataset. For example, the compound $TeO_2$ was decomposed into its constituent elements Te and O to form our training set.,

**Binary Compound Structure Relaxation Task (CrR)**

- We selected 2,000 binary compound crystal structures from the Materials Project (Jain et al., 2013) across the following categories: Intermetallics, Semiconductors, Oxides, Sulfides, Nitrides, Carbides, Hydrides, Halides, Borides, Silicides, Phosphides, Arsenides, Tellurides, and Selenides. And we applied perturbations to alter the positions of certain atoms and modify the cell parameters of these structures to form our training dataset.

### E.4 RESULTING DATA MIXTURE

The pretraining dataset was post-processed using an annotation pipeline to detect each molecule in the texts. For each molecule, the tags "[START_MOL]" and "[END_MOL]" were added to enclose it. Similarly, the SMILES were computed for each molecule and added between "[START_SMILES]" and "[END_SMILES]" tags after the molecule.

Table 16: MiST Pretraining Dataset Composition

| Data Source | Tokens | Proportion |
|---|---|---|
| ChemRxiv + S2ORC | 1.2B | 41.37% |
| FineWeb (Q4–6) | 1.4B | 48.27% |
| PubChem Synthetic | 120M | 4.14% |
| Synthetic Reactions | 100M | 3.44% |
| CommonCrawl Replay | 80M | 2.75% |
| **Total** | **2.9B** | 100% |

Supervised fine-tuning was performed on the MiST - Qwen-3B model, primarily using chemistry-specific reasoning and instruction datasets, as follows:

Table 17: MiST SFT Dataset Composition

| Data Source | Contents/Size |
|---|---|
| DeepSeek Rxn Traces | ~7,000 samples |
| SmolInstruct | I2S, S2I, captioning, gen. |
| MMLU | 350 general + 300 chemistry samples |
| Chain-of-Thought (CoT) | ~27,000 samples |

# F  COMPUTE RESOURCES

As described in Section D.2 for the GRPO experiments, each training was run for 12 hours on four nodes (with 4 NVIDIA GH200 120GB GPUs or 8 AMD MI250x 128GB GPUs), summing to 16 GPUs and 192 GPU-hours per training. The best hyperparameters and rewards were optimized using a total of 30k GPU-hours with variations in the experimental setups. An additional 10k GPU-hours were used for the final runs, summing to a total of 40k GPU-hours.

# G ADDITIONAL EXPERIMENTAL RESULTS

## G.1 MIST

We conducted other experiments to evaluate our MiST model's performance on other tasks and in comparison with strong baselines from the literature. In particular, we compare against NatureLM Xia et al. (2025) and other general-purpose LLMS, on the task of SMILES to IUPAC and IUPAC to SMILES conversion. The results shown below put our MiST model (3B) on par with NatureLM 8B, while approaching the 8x7B MoE variant on IUPAC-to-SMILES conversion.

Table 18: Accuracy for IUPAC-to-SMILES and SMILES-to-IUPAC on benchmark datasets. The best value in each column is shown in bold.

| Model | IUPAC-to-SMILES | SMILES-to-IUPAC |
|---|---|---|
| STOUT | 0.735 | 0.565 |
| GPT-4 | 0.033 | 0.0 |
| Claude 3 Opus | 0.177 | 0.0 |
| LlaSMol_Mistral | 0.701 | 0.29 |
| NatureLM (1B) | 0.476 | 0.284 |
| NatureLM (8B) | 0.679 | 0.517 |
| *Qwen+MiST+SFT* | 0.682 | 0.445 |

## G.2 RL

From Table 19, it can be observed that the base model, Qwen-2.5 3B, possesses a degree of domain knowledge in materials science sufficient to generate some valid compositions. However, the relatively low scores suggest that the model is primarily retrieving compositions seen during training or generating valid combinations through rough heuristics. This is further supported by its low SCS, which indicates a limited understanding of compositions at the symbolic level.

The introduction of MiST leads to a significant improvement in SCS, as MiST specifically targets symbolic competence during training. However, since the model was not trained directly on materials science data and has a relatively small parameter size, it likely replaced some of its prior knowledge with representations more aligned with SMILES syntax. This shift contributes to the lower validity and precision scores, reflecting a reduced ability to follow instructions in non-SMILES-based tasks. As a result, the model often fails to generate outputs in the required format, especially when it encounters ambiguous prompts or reaches its maximum output length.

Fine-tuning the MiST model using SFT yields improvements in both SCS and instruction-following ability, as evidenced by higher validity and precision scores. These gains suggest that the model is able to recover some materials science knowledge while refining its symbolic understanding. However, the low novelty score indicates limited generalization, implying that the model is overfitting to training data and struggles to produce truly new compositions.

In comparison, SFT applied directly to the base Qwen-2.5 3B model results in high validity and precision but retains a poor SCS score. This contrast highlights that symbolic competence is primarily achieved through MiST, not SFT. Additionally, the low novelty score again suggests overfitting, as the model continues to rely on memorized examples rather than generating original compositions.

When combining MiST, SFT, and RL, there is a substantial improvement in novelty, indicating that the model is better able to utilize its symbolic understanding and domain knowledge to generate rather than recall compositions. This suggests that while base models have weak symbolic competence, MiST significantly enhances this capability. Though MiST initially reduces instruction-following ability due to longer and more complex outputs, SFT helps regain this ability for specific tasks. Ultimately, RL fine-tuning balances symbolic competence with domain-specific generation, enabling the model to produce valid, precise, and novel compositions using the specified elements.

In contrast to the findings observed in the Conditional Material Generation task, we did not detect any notable improvement in CCS after introducing MiST to the Binary Crystal Structure Relaxation task. This discrepancy arises because the Binary Crystal Structure Relaxation task specifically emphasizes

Table 19: CMG = Conditional Material Generation.

| Model | SCS ↑ | CCS ↑ | Validity ↑ | Precision ↑ | Novelty ↑ |
|---|---|---|---|---|---|
| **Qwen-2.5 3B** | 0.122 | 0.828 | 58.6 | 68 | 74.8 |
| +MiST | 0.989 | 0.795 | 1.2 | 0.67 | 84.6 |
| +SFT | 1.142 | 0.785 | 34.8 | 38.5 | 49.2 |
| +RL | 0.893 | 0.777 | 73.8 | 97.1 | 91.3 |
| **Ablations** | | | | | |
| no MiST + SFT | 0.199 | 0.824 | 87.4 | 93.9 | 60.2 |

Table 20: CrR = Binary crystal stucture relaxation, CeB = Chemical formula balancing.

| Model | SCS ↑ | CCS ↑ | CrR ↑ | CeB ↑ |
|---|---|---|---|---|
| **Qwen-2.5 3B** | 0.346 | 0.834 | 0 | 1.2 |
| +MiST | 0.355 | 0.795 | 0 | 26 |
| +SFT | 0.528 | 2.361 | 16.2 | 29.2 |
| **MatSci Tasks** | | | | |
| +RL(CrR) | 0.447 | 2.599 | 65 | — |
| +RL(CeB) | 1.653 | 0.666 | — | 47 |
| **Ablations** | | | | |
| no MiST + SFT(CrR) | 0.573 | 2.652 | 12.6 | – |
| no MiST + SFT(CeB) | 1.494 | 0.849 | — | 45 |

structural relaxation, a domain not directly targeted by MiST training. Consequently, MiST did not enhance the model's chemical competence related to structural relaxation.

However, subsequent fine-tuning via SFT successfully incorporated relevant domain knowledge into the model, resulting in substantial performance improvements on the task. This step notably increased the model's capability to accurately execute structural relaxations, which was previously limited. Moreover, further refinement through reinforcement learning (RL) effectively enhanced the model's success rate, demonstrating that the integration of RL optimally balances domain-specific expertise with task-oriented performance improvements.

We further conducted an additional analysis across all 200 test set datapoints, and observed that the model performed comparably across the five crystal systems included in the test set.

Table 21: Summary of Crystal Systems for the MiST + SFT + RL (CrR) Model. This table presents a detailed breakdown of the performance (accuracy) of the MiST + SFT + RL (CrR) task, as shown in the Table, evaluated separately across different crystal systems.

| Crystal System | Average Accuracy | Total Samples |
|---|---|---|
| Tetragonal system | 0.6383 | 47 |
| Orthorhombic system | 0.6897 | 29 |
| Hexagonal system | 0.6250 | 72 |
| Trigonal system | 0.6572 | 35 |
| Monoclinic system | 0.7143 | 7 |
| Cubic system | N/A | N/A |
| Triclinic system | N/A | N/A |

We illustrate the capability of our Mist + SFT + RL model to reduce the inner energy of a perturbed, unstable ZnSe-P4_nmm crystal structure within 10 steps, where the stable state of the ZnSe-P4_nmm crystal has an inner energy of -2.94069766998291.

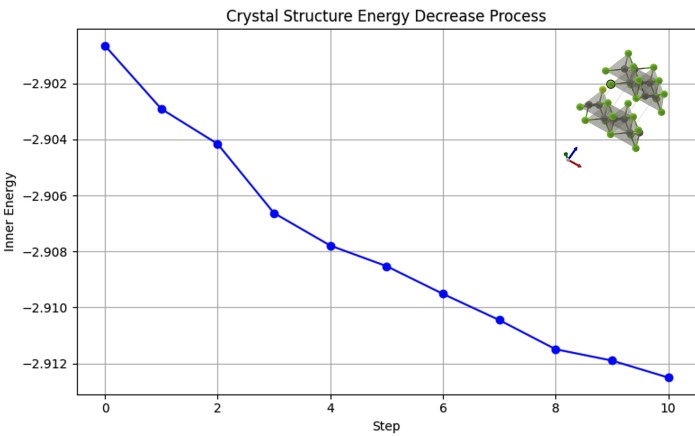

Figure 12: Graph demonstrating the relaxation of the ZnSe-P4_nmm crystal structure with the Mist + SFT + RL model

## H  MODEL ANALYSIS

Here, we provide some analysis of the models' behavior. For the following example reaction:

**Question**
```
Reason and predict the correct product in
SMILES notation from the following reaction
CC(C)(C)C(=O)c1oc2nc(-c3ccc(Cl)cc3Cl)c(-c3ccc(Cl)cc3)cc2c1N.CCBr
```

**Correct answer**

```
CCNc1c(C(=O)C(C)(C)C)oc2nc(-c3ccc(Cl)cc3Cl)c(-c3ccc(Cl)cc3)cc12
```

After RL training, although the output format is correct, the base model Qwen2.5-3B (+RL) fails to predict the correct product of this reaction, with simple reasoning and incorrect reaction type (esterification reaction). On the other hand, the variant with SFT exhibits a more detailed reasoning that delves into the functional groups of the precursors. However, it hallucinates the presence of sodium hydride and the hydroxyl group, which are not present in the input reagents, leading to an incorrect result. This could be attributed to the lack of SMILES symbolic understanding. Lastly, the model that underwent the full pipeline (MiST + SFT + RL) successfully reasons the correct product with reasonable description of the reactants' structure and mechanism, showing better symbolic understanding and Chemical knowledge.

The full responses of these models are given below:

---
**Qwen2.5-3B + RL [Wrong]**
---

```
<think> The given reactants are a carboxylic acid and a
bromoalkane.  The reaction between these two compounds is
an esterification reaction, where the carboxylic acid
reacts with the alkyl halide to form an ester and a halide.
The product of this reaction is an ester with the general
formula R-COOR'. In this case, the product is a carboxylic
acid ester with the general formula R-COOR', where R is the
alkyl group and R' is the alkyl group from the bromoalkane.
</think>
<answer> COC(C)C(O)C(C)C(=O)OCC1CCCCC1 </answer>
```

**Qwen2.5-3B + SFT + RL [Wrong]**

<think> The reaction involves the alkylation of a phenolic oxygen in the first compound using ethyl bromide CCBr( CCBr) under basic conditions. Here's the step-by-step mechanism:
1. Deprotonation**: Sodium hydride [H-].[Na+]( [Na+][H-]) deprotonates the hydroxyl group(- OH [OH-]) on the benzene c1ccccc1 ring of the first compound( CC(C)(C)C(=O)c1oc2nc(-c3ccc(Cl)cc3Cl)c(-c3ccc(Cl)cc3)cc2c1N), generating a phenoxide [O-]c1ccccc1 ion. The phenoxide [O-]c1ccccc1 is a strong nucleophile.
2. Nucleophilic Substitution(SN2)**: The phenoxide [O-]c1ccccc1 ion attacks the electrophilic carbon [C] in ethyl bromide CCBr( CCBr). The bromide [Br-] ion( Br) is displaced, forming a new ether CCOCC bond( C-O-C) between the benzene c1ccccc1 ring and the ethyl group.
3. Product Formation**: The final product retains the original fused aromatic system of the first compound, with the hydroxyl group replaced by an ether CCOCC-linked ethyl group. The tert-butyl carbonyl( CC(C)(C)C(=O)), oxazole c1cocn1 ring(oc2...), and amine group(N) remain unchanged.
Product SMILES**: `CCOCc1ccc(-c2cc(-c3cc(Cl)cc(Cl)c3)c(N)c(=O)c2OC)cc1 `
Explanation**: CCOC Ethyl ether CCOCC group formed from ethyl bromide CCBr. c1ccc(...)cc1**: Benzene c1ccccc1 ring from the first compound. c2cc(...)c(N)...**: Fused oxazole c1cocn1 ring(positions 1-6) with substituents retained. c3cc(Cl)cc(Cl)c3**: Chlorinated benzene substituent from the first compound. N Amine group preserved in the fused ring. OC Oxazole c1cocn1 ring(positions 2-5) with a hydroxyl group replaced by the ether CCOCC linkage.
This reaction exemplifies a Williamson ether synthesis, where a phenoxide [O-]c1ccccc1 displaces a halide in the presence of a strong base to form an ether CCOCC. </think>
<answer>
CCOCc1ccc(-c2cc(-c3cc(Cl)cc(Cl)c3)c(N)c(=O)c2OC)cc1
</answer>

**Qwen2.5-3B + MiST + SFT + RL [Correct]**

<think> The reaction involves the alkylation of a primary
amine group in the first molecule by bromoethane CCBr(
CCBr).  Here's the step-by-step mechanism:
1.   Reactant Analysis**:
First molecule**:  A bicyclic compound with a tert-butyl
group( CC(C)(C)C(=O)), a furan c1ccoc1-like ring(
c1oc2nc(...)cc2c1N), and two chlorobenzene Clc1ccccc1
substituents( c3ccc(Cl)cc3 and c4ccc(Cl)cc4).
Second molecule**:  Bromoethane CCBr( CCBr), where the
bromine [Br] is a good leaving group.
2.   Mechanism**:
The lone pair on the primary amine nitrogen [N]( N [N]) in
the first molecule acts as a nucleophile.  This nitrogen
[N] undergoes an SN2 reaction** with bromoethane CCBr.  The
amine attacks the electrophilic carbon [C] adjacent to the
bromine [Br] in bromoethane CCBr, displacing the bromide
[Br-] ion( Br).  The result is the formation of a secondary
amine** in the first molecule, where the nitrogen [N] is
now bonded to an ethyl group( CC).
3.   Product Structure**:
The nitrogen [N] in the furan c1ccoc1-like ring is replaced
by a CH2CH3 group, making it NCC** in the product.  All
other substituents( tert-butyl, chlorobenzenes [C-]#CCl,
and furan c1ccoc1 ring) remain unchanged.
Final Answer**:
The product SMILES is
CC(C)(C)C(=O)c1oc2nc(-c3ccc(Cl)cc3Cl)c(-c3ccc(Cl)cc3)cc2c1NCC`.
The reaction selectively alkylates the primary amine group
in the bicyclic compound with bromoethane CCBr, converting
it to a secondary amine while retaining the rest of the
structure.  </think>
<answer>
CC(C)(C)C(=O)c1oc2nc(-c3ccc(Cl)cc3Cl)c(-c3ccc(Cl)cc3)cc2c1NCC
</answer>

## I  REPRODUCIBILITY

All the code used to produce the results presented in this work can be found under `https://figshare.com/s/7afa3c106d6d874e4094`. The continued pretraining and supervised fine-tuning, as described in Section 4 and Appendix D, have been conducted using the nanotron library (see `https://github.com/huggingface/nanotron`). The configuration files and datasets used are released at `https://figshare.com/s/7afa3c106d6d874e4094`.

## TABLE OF RELEASED ASSETS

| Asset | Usage Instructions | License/Citation Info | Location/URL |
|---|---|---|---|
| Source code | Download and unzip. See `README.md` for installation and experiment scripts (`run_train.py`). | MIT License. Please cite this paper. | `https://figshare.com/s/7afa3c106d6d874e4094` |
| Model checkpoints | Download the archive. Full instructions in `README.md`. | MIT License. Please cite this paper. | `https://figshare.com/s/7afa3c106d6d874e4094` |
| Datasets (pretraining/fine-tuning splits) | Download files; load as a HuggingFace Dataset. | For research use only. Cite the original dataset and this paper. | `https://figshare.com/s/7afa3c106d6d874e4094` |
| Training configs | Config YAML files for nanotron available as `.yaml`; pass as argument to Nanotron CLI. | MIT License. | `https://figshare.com/s/7afa3c106d6d874e4094` |

Table 22: List of digital assets released with this work, including usage instructions and licensing/citation information. Note: All assets are hosted anonymously on Figshare for double-blind review.

All digital assets (code, models, data splits, and configs) are provided through anonymous Figshare links for double-blind review, as recommended by NeurIPS guidelines. After publication, these will be migrated to a permanent repository.

