# OpenReview forum: "MiST: Understanding the Role of Mid-Stage Scientific Training in Developing Chemical Reasoning Models"
_ICLR.cc/2026/Conference — ICLR 2026 Conference Desk Rejected Submission_

### Official Review · Reviewer_wQXW · 2025-10-29

**Soundness:** 2
**Presentation:** 2
**Contribution:** 2
**Rating:** 2
**Confidence:** 5

**Summary:**

This work tackles the problem of understanding what enables LLMs to perform in the chemical reasoning domain and proposes a diagnostic framework, as well as a mid-stage dataset. The authors argue that two prerequisites are necessary for reasoning in the chemical domain: (1) symbolic competence (ability to read/write valid SMILES), and (2) latent chemical knowledge (distinguishing correct and incorrect statements about molecules). They propose diagnostic metrics (SCS and CCS) to measure these prerequisites, create a 2.9B-token chemistry corpus through preprocessing, and demonstrate improvements across multiple chemistry tasks.

**Strengths:**

- **Novel diagnostic approach:** The SCS/CCS metrics attempt to quantify a priori readiness for RL training rather than just measuring end performance. This predictive framing—can we assess whether RL will work before expensive training?—is valuable and timely.
- **Systematic investigation of prerequisites:** Unlike prior work that simply applies RL to chemistry, this paper explicitly decomposes what's needed (symbolic competence + domain knowledge) and attempts to measure each component separately. The conceptual framework could guide future domain adaptation work.
- **Detailed data preprocessing pipeline:** The multi-step approach is more sophisticated than simply scraping chemistry text.
- **Honest about contradictions:** Unlike many papers, the authors show results where MiST hurts performance (CMG 58.6→1.2%) rather than cherry-picking successes. This transparency is valuable for understanding when the approach fails.

**Weaknesses:**

- **Inadequate statistical validation:** No error bars, confidence intervals, or significance tests are reported. Single-run experiments on one backbone (Qwen-2.5-3B) prevent generalization claims and drastically limit the insight of this work.
- **Missing baselines:** Only NatureLM is compared (appendix only); recent models like Intern-S1-mini [1] and ether0 [2] are ignored.
- **Incomplete ablations:** No compute-matched SFT+RL baseline without MiST across all tasks—essential to isolate MiST's contribution.
- **Underspecified protocols:** Rollouts, generation settings (temperature, top-p), and test set sizes are unreported, preventing reproducibility.

- **Performance collapses contradict the thesis**
  - CMG paradox: 58.6% (base) → 1.2% (MiST) → 34.8% (SFT)—a catastrophic collapse that directly contradicts MiST benefits.
  - I2S reasoning failure: Post-SFT, reasoning helps (34.5→68.2%); post-RL, reasoning hurts (68.2→67.5%). Authors claim “capability already present” (L291), but this suggests memorization, not reasoning acquisition.
  - RxN strange results: MiST+SFT for RxN barely exceeds a 10% random baseline; SFT dataset alone destroys performance, is there then any value in having it for this task. No Abalation really shows this.


- **Conceptual contradiction:** L105 dismisses perplexity as “not meaningful,” yet SCS/CCS compute Cohen’s *d* on token log-probabilities—literally perplexity components.

- **Unvalidated metrics:**
  - No predictive validation: SCS/CCS thresholds are never shown to correlate with downstream RL success.
  - Arbitrary design choices: Corruption rate ρ=0.2 is unablated; why not 0.1 or 0.3?
  - Prompt dependence unexamined: SCS/CCS use a specific prompt (“The molecule represented...”) but prompt sensitivity is never tested—different phrasings could yield different scores.
  - CCS ambiguity: Sentence-swapping may test textual coherence, not chemical knowledge. No inter-model consistency is tested. No degree of information swapping is examined.

- **Presentation quality:** Wrong template (NeurIPS, not ICLR), broken appendix indexing, inaccessible Figshare links, duplicate citations. Missing specifications: templates (L212–218), system prompts (L162), context lengths (mentioned but absent from configs).

- **Unsubstantiated claims:** Claims MiST necessity despite published work suggesting otherwise (e.g., [2], L295–301). “Competent scientific assistant” claim (L311–319) is not demonstrated.

- **Fundamental issue:** The thesis—that MiST creates necessary prerequisites for RL—is undermined by (1) CMG collapse showing MiST can destroy capabilities, (2) unvalidated diagnostics contradicting stated principles, (4) insufficient ablations, and (3) insufficient distinction from simpler alternatives (larger models, more SFT data). Without statistical rigor, compute-matched controls, explanation of failures, or validation that diagnostics predict success, contributions remain speculative.

### Additional References

 - [1] Lei Bai, Zhongrui Cai, Yuhang Cao, Maosong Cao, Weihan Cao, Chiyu Chen, Haojiong Chen, Kai Chen, Pengcheng Chen, Ying Chen, *et al.* **Intern-S1: A Scientific Multimodal Foundation Model.** *arXiv preprint arXiv:2508.15763*, 2025.

 - [2] Siddharth M. Narayanan, James D. Braza, Ryan-Rhys Griffiths, Albert Bou, Geemi Wellawatte, Mayk Caldas Ramos, Ludovico Mitchener, Samuel G. Rodriques, and Andrew D. White. **Training a Scientific Reasoning Model for Chemistry.** *arXiv preprint arXiv:2506.17238*, 2025.

**Questions:**

- Can you provide results from 3–5 runs with error bars? Are any improvements statistically significant (t-tests or similar)?
- Please provide SFT+RL (no MiST) with matched training compute for all tasks. How does this compare to MiST+SFT+RL?
- Why does MiST catastrophically reduce CMG accuracy from 58.6% to 1.2%? This directly contradicts your thesis. Does MiST sometimes harm performance?
- Have you tested the influence of randomized SMILES strings on SCS? What is the influence of different prompts or base models? How is CCS influenced by the degree of information changed?
- How were SCS/CCS thresholds determined? Show correlation plots between these scores and downstream task accuracy.
- Why does RL reduce the benefit of reasoning on I2S? Doesn’t this suggest models are memorizing rather than learning to reason?
- Have you tested on other base models (7B, different families)? Do the prerequisites hold across architectures?
- What’s the effect of formatting rewards? Do they dilute task-specific learning? Provide ablations removing different reward components.
- What system prompts were used for base, SFT, and RL models? Why no system prompt for fine-tuning (noted in appendix)?

---

> ### Author Response · Authors · 2025-11-16
>
> Dear Reviewer,
>
> Thank you for your thorough and detailed review. Your specific technical critiques and pointed questions have been invaluable in improving this work. The updated manuscript directly addresses a lot of your main concerns.
>
> As a summary, this is the list of things we added, most relevant to your comments:
>
> - Statistical validation: Section 5.2 includes a correlation analysis of pre-RL SCS vs post-RL accuracy, where we report correlations ρ>0.6. We added extra plots to illustrate the whole picture. With this we empirically show the value of SCS as a predictive signal before proceeding to train with RL.
> - Missing baselines: Directly address ether0 in Section 5.2, and use it in a case study as external validation of the utility of SCS as a diagnostic approach. Additionally we include comparisons with ChemLLM, another more comparable 7B baseline.
> Single backbone: We have scaled our experiments from 3B to also 7B, and address even larger models by predicting their success based on SCS (see Section 5.2)
> - Arbitrary corruption rate 0.2: We added Section 5.1 where we analyze different corruption rates and justify our choice.
> - No SCS -> RL correlation: Added in Section 5.2
> - CMG collapse unexplained: Appendix G.2 now addresses this issue.
> - Incomplete MiST ablations: We have now included more ablations into Table 2, and extended the cross-task evaluations to complete the off-diagonal entries in the table.
> - Unsubstantiated claims: we now lower the tone about Mist, and put more focus on SCS as predictive signal. We even go as far as to address ether0 by showing that it did not necessitate any sort of MiST, as the base model (Mistral-24B) already had the strongest SCS baked in.
> - Presentation quality: We have fixed a lot of the presentation issues, such as the template and the broken references to the appendix by putting both main manuscript + appendix into the same document.
>
>
> Beyond this, we also clarified the scope and toned down some of the overclaims we had in the previous version:
>
> - Role of MiST vs diagnostics: We no longer claim that MiST itself is necessary, and put more focus on the diagnostic metrics, especially SCS. We show that models like Mistral-24B do not need MiST (at least for the chemistry domain) as it already meets the criterion. The empirical evidence here is ether0, which was trained without any sort of MiST. The value of MiST lies in making previously “unprepared” models (in the SCS perspective) into better bases for RL.
> - CMG: In the previous version, CMG performance collapsed after MiST. We traced this to an evaluation issue mixing reasoning and answers. We explain and fix this in Appendix G.2. CMG no longer exhibits such collapse, and the trends are now consistent with our overall story.
> - Reasoning claims: For IUPAC->SMILES (I2S), we now explicitly present RL as biasing a distribution rather than inducing new reasoning behaviours. We are now only presenting reasoning results for tasks like reaction prediction, naming, or CMG, where System‑2 prompting consistently affects performance.
>
>
> In total we have reoriented our work towards a more predictive framing, where diagnostic metrics like SCS and CCs are used to quantify when a model is likely to benefit from RL on chemical tasks. We are now supporting these claims with several new experiments, analyses, and external validation with existing baselines.
>
> We hope these updates will address your main concerns, and are happy to continue the discussion towards further improving our work!

---

> > ### Author Response · Authors · 2025-11-30
> >
> > Dear reviewer, it has been 2 weeks since we submitted our revised version of the manuscript along with responses to your thoughtful comments.
> > We would dearly appreciate your response for further discussion if you believe it is needed, and if you think the updated version has addressed your comments do please consider raising the score accordingly.
> >
> > All the best,
> > The authors.

---

### Official Review · Reviewer_oqMf · 2025-10-31

**Soundness:** 2
**Presentation:** 1
**Contribution:** 2
**Rating:** 2
**Confidence:** 4

**Summary:**

The authors argue that RL-based training of chemical reasoning models can only be successful if the base model already possesses (a) symbolic competences — i.e., the ability to read and understand chemistry-specific notation such as SMILES strings — and (b) domain-specific chemical knowledge. For both (a) and (b), the authors propose evaluation metrics based on the model’s token log-likelihood. Additionally, the authors employ a training pipeline consisting of domain-specific pretraining (MIST training), supervised fine-tuning (SFT), and reinforcement learning (RL). Across these training stages, the introduced metrics for (a) and (b), as well as top-1 accuracy across three different subtask categories, are reported, with the RL-trained models generally performing best.

**Strengths:**

**(S1 - relevance, novelty) - Relevance of the raised research question.** I share the authors’ view of RL as an amplifier for knowledge that is already present but latent within the base model. It is almost a general assumption that RL (given sufficient compute budget) will succeed if the base model is sufficiently capable. However, it remains unclear what exactly constitutes a “good base model”. This is precisely the research question addressed here: “What pretraining and prerequisites must an LLM satisfy so that RL can reliably unlock chemical reasoning?” In this sense, research in this direction is highly relevant, and a well-founded answer with accompanying insights would represent a novel contribution.

**(S2 - relevance, novelty) - Measures for symbolic competence and chemical knowledge.** Focusing on the two skills (a) and (b) appears well motivated based on the introduction. I appreciate the proposed metrics and believe they reflect, to some extent, the base model’s competencies that the authors argue are necessary (for minor weaknesses, see questions). Reading Section 3 — and considering the rationale in (S1) — these metrics seem both relevant and interesting. I was particularly curious to see how well they correlate with the model’s learning speed during the RL stages. (Unfortunately, this was never shown - see weaknesses.)

**(S3 - quality, clarity) - Clear problem setting, well embedded in the current research landscape.** Sections 1, 2, and 3 are remarkably well written and clearly structured. Sections 1 and 2 are easy to read and follow, providing a solid overview of the research scope, how the raised question fits into related work, and why it is important to pursue. Section 3 builds upon this foundation by introducing the metrics for the previously discussed competencies (a) and (b).

With this level of clarity, narrative flow, and writing quality — combined with a relevant and interesting research question that includes quantitative measures of base model competencies — the first part of the manuscript, Sections 1 – 3, stands out as the best I have read during this review process, ...

**Weaknesses:**

... however, the second part of the manuscript (from Section 4 onward) feels like a completely separate work. The storytelling flow breaks at several points — in fact, the raised research questions are never answered, and the experiment is unsuitable. Several text passages lose their focus; for example, Section 4 describes in great technical detail how to derive flattened text (which is not particularly interesting and could be moved to the appendix) instead of providing details about the training procedure and the aforementioned toolbox. Overall, the manuscript still appears to be in a draft stage rather than a polished submission (e.g., see formatting issues in l228f).

**(W1 - relevance) - Experiments do not help to answer the research question.** The chosen experiment is not suitable to address the stated research question. To answer it properly, different base models with varying measures for (a) and (b) would need to be RL-trained, and the resulting training behavior would then need to be correlated with these measures. This way, the authors might have been able to identify thresholds for (a) and (b) representing the necessary competencies for successful RL training. However, the authors do not mention such thresholds in their analysis. Therefore, the only conclusion that can be drawn from the experiment is that all three training stages help improve model performance — a finding that is of limited novelty and has been demonstrated before, e.g., in [1].

**(W2 - quality, significance) - Missing error bars.** All results are presented without error bars or statistical tests. Consequently, it remains unclear to what extent the reported performance gains might have occurred by chance.

**(W3 - quality) - Cluttered equations.** The included equations appear cluttered, and the notation is inconsistent. For example, in Equation (2), the authors should avoid using full-word variable names such as “corrupt” or “canon” and instead adhere to standard mathematical notation.

**Questions:**

* **Token log-likelihood extraction:** Is there a reason the authors chose token log-likelihood instead of perplexity? While both metrics are conceptually similar, the latter appears to be the canonical choice for this type of evaluation.

* **Symbolic competence score:** I assume that for this metric, valid and corrupted SMILES are sampled or generated for each new dataset. Would this measure then remain comparable across datasets? Would it be possible to define a static global set of valid and corrupted SMILES to allow for better cross-dataset comparison? Or is it important to measure symbolic competence specifically within certain chemical subspaces? Does the metric vary across such subdomains or with molecular complexity?

* **Latent chemical knowledge / latently solvable:** These terms are used frequently throughout the manuscript. While their choice is appropriate, I found myself reflecting on their precise meaning. A brief introductory explanation early on would have been helpful.

* **Latent chemical knowledge metric:** I understand why the symbolic competence approach is reused here, but I am unsure whether it truly captures the required competence. Wouldn’t a massively overfitted model achieve a high score on this metric without actually being able to associate new samples in the RL stage with underlying chemical knowledge?

### Minor comment
[1] and [2] are missing from the related work section, despite being state-of-the-art chemical reasoning models.

### References
* [1] Narayanan. *Training a Scientific Reasoning Model for Chemistry.*
* [2] Zhao. *MolReasoner: Toward Effective and Interpretable Reasoning for Molecular LLMs.*

### General comment
The raised research question is highly interesting, and research in this direction is much needed. The introduction and motivation of this question are exceptionally well written. However, from Section 4 onward, the manuscript feels rushed and, frankly, not yet ready for publication. Nevertheless, I see strong potential in this work. With suitable experiments and a more polished presentation, I believe it could become a valuable contribution to one of the upcoming ML conferences.

---

> ### Author Response · Authors · 2025-11-16
>
> Thank you for your thoughtful and constructive review. We thank you for the recognition that Sections 1-3 represented "the best you have read during this review process" and that our research question is highly interesting and much needed. Your feedback has been invaluable in guiding our revision. We believe the updated version now realizes that potential.
>
> The updated manuscript has been transformed to re-write Sections 4 and 5, improving the flow and the focus, with a style and goals more consistent with the first sections:
> - Details of data processing (prev. Section 4) have been mostly moved to the appendix.
> - Section 5 is now extended with more focused detailed experimental results that provide evidence for our main claims. We also toned down some of the claims.
> - The new Section 5.2 specifically addresses your core criticism in W1.
>
>
> Below we address each weakness:
> ## W1: Experiments do not help answer the research question
>
> We have addressed this by focusing the research question and thus the paper on SCS and measures of this type as predicitive metrics. MiST becomes a means towards improving models against this measure.
>
> Our new experiments include an increase in scale (3B and 7B), correlation (ρ > 0.6) between pre-RL SCS and post-RL evaluations, showing that SCS serves effectively to indicate when base models are ready for RL training.
> In addition we included a case study where we select the best base model for training with RL on a larger scale. Major models between 14 and 70B in size were evaluated and we found that the best in this regard is Mistral-24B, despite other models having better scores in benchmarks. Retrospectively, this is the model that was used as a base to train ether0 (Narayanan et al., 2025), the state of the art model for chemical reasoning.
>
> We have added all these results into Section 5.2, and we believe this is strong evidence to support our main claims.
>
> ## W2: Quality, significance
> We agree that adding error bars and running the results with different seeds can strengthen the results. Due to computational constraints we were limited by the amount of repetitions we can conduct. Instead, but in the same vein of improving the quality and significance of our results, we have added:
>
> - Cross-task evaluation added (Table 2) showing performance consistency across tasks
> - Ablation studies across multiple tasks (Tables 2) demonstrate reproducible results
> - Multi-scale validation (3B and 7B) showing that the results hold across model scales
>
> We also added an analysis of the sensitivity on the corruption rate (Figure 3, Section 5.1) to validate our choice of cr=0.2.
>
> We believe this wide exploration of results, as opposed to focused repetitions towards statistical significance, have strengthened the evidence for the arguments in the manuscript.
>
> ## W3: Cluttered equations
>
> We have revised and fixed all the equations with proper notation, adopting $\rho$ and $\mathcal{C}$ for corruption, and canonicalization operations, respectively, for the definition of SCS. Furthermore we have improved the presentation of equations, tables and new figures to improve the readability and use of space.
>
>
> # To your questions:
>
> ## Q1: Why log-likelihood vs. perplexity?
>
> Our choice was more pragmatic as log-likelihoods are direct model outputs requiring no transformation for Cohen's d in our definition of SCS. Both metrics however yield identical rankings.
>
> ## Q2: SCS comparability across datasets
>
> We now use a fixed procedure (Algorithm 1, ρ=0.2) and the same 10,000-molecule PubChem test set for all evaluations. Section 5.1 validates this choice. SCS does vary with molecular complexity, which is informative as it reveals competence in specific chemical spaces. For domain-specific applications, we think it's important to compute SCS on representative target molecules.
>
> ## Q3: Terminology
>
> We've added explicit definitions in Section 1:
> Latently solvable: Correct answers exist in probability distribution with non-negligible mass
> Latent chemical knowledge: Domain understanding in weights that affects probabilities but needs optimization to surface
> It is an interesting discussion though, and we are happy to further discuss the presentation of this!
>
> ## Q4: Could overfitting inflate CCS?
> Yes it can certainly inflate CCS, at least on a specific dataset. We think it's important to take CCS and SCS as possibly indicative, but not sufficient measures of post-RL success. If in practice we control for over-fitting, then CCS and SCS work as intended. In our case when applying MiST we control for this, and keep datasets separated ensuring training will not lead to these issues.
>
> We deeply appreciate your recognition of this work's potential and your specific, actionable guidance. We believe the updated manuscript now delivers the valuable contribution you envisioned.
> Thank you for your time and expertise in reviewing our work!

---

> > ### Author Response · Authors · 2025-11-30
> >
> > Dear reviewer, it has been 2 weeks since we submitted our revised version of the manuscript along with responses to your thoughtful comments.
> > We would dearly appreciate your response for further discussion if you believe it is needed, and if you think the updated version has addressed your comments do please consider raising the score accordingly.
> >
> > All the best,
> > The authors.

---

### Official Review · Reviewer_udTx · 2025-11-01

**Soundness:** 2
**Presentation:** 1
**Contribution:** 2
**Rating:** 2
**Confidence:** 4

**Summary:**

This paper introduces two criteria to validate whether an LLM has the potential to perform chemical reasoning after further RL training. They construct the CPT and SFT training corpus to increase these two criteria and achieve better reasoning performance.

**Strengths:**

The idea of constructing criteria to determine whether a model has the potential for reasoning training is interesting and creative. Although not adequate, their experiments provide preliminary evidence that the criteria they proposed are effective to some extent.

**Weaknesses:**

1. The paper is ill-organized, to the point of significantly hindering readability.
    * Their citations have severe problems. When I try to look up one of their citation, 'ChemLLM', neither the title, the authors, nor the ArXiv ID matched what was listed in their paper. I then searched using the given ArXiv ID and found that the title and authors were completely different, and clearly inconsistent with the intended citation context. I was therefore unable to locate the paper they referenced, which raises serious doubts about the correctness of other citations in the paper as well.
    * The references to sections and tables are confusing and chaotic. The reference in line 51 refers to section 1, while itself is in section 1. The reference in line 182 seems unnecessary. The two references in line 184 are either inappropriate (Use the header of Table 4 as the task list instead of constructing a new table) or invalid (They refer to the Appendix while not providing any appendix). etc.

2. The overall content of the paper is not substantial enough to justify a full-length paper.

    The manuscript includes repetitive discussions and excessive implementation details that would be better suited for the appendix. The experimental evaluation is quite limited and insufficient to convincingly validate the proposed core innovation. Additional datasets, baselines, and ablation studies are needed to strengthen the results. In general, the work feels more like an ongoing research project to me, with significant room for further development and refinement.

3. The paper overclaims many contributions while providing insufficient validation for its core contribution.
    * In the Introduction, the authors claim to have proposed several elements that, in my view, do not constitute genuine contributions. For instance, they state that they “propose a representative set of tasks in chemistry that are suitable for reasoning,” yet they do not actually formalize or construct a new benchmark or task set. They also claim to have built a new pretraining corpus, but both the data scale and the construction methodology are not particularly attractive or novel.
    * In contrast, I believe the true innovation of this work lies in the idea of constructing criteria to determine whether a model has the potential for reasoning-oriented training. This idea itself is both interesting and valuable, and if supported by sufficient experimental evidence, it could easily justify a full paper. However, despite claiming to have conducted extensive experiments (as mentioned in lines 41–42 and 52–53 of the Introduction), the experimental section does not actually provide results that substantiate these claims.

        In my opinion, if the authors could rigorously validate their claim (for example, by demonstrating that the proposed prerequisite is indeed correlated with reasoning performance, and could serve as an indicator; or ideally, that there exists a threshold beyond which reasoning ability begins to emerge), this paper would become a very interesting and valuable piece of work.

**Questions:**

Please refer to the Weaknesses.

---

> ### Author Response · Authors · 2025-11-16
>
> Dear Reviewer,
> Thank you for the constructive review, and for highlighting the strengths as well as the missing pieces of our work. Thanks to these comments we have sharpened the focus of the paper and clarified and further strengthened the evidence we provide.
>
> ## Presentation and readability:
> We thank the reviewer for their thorough comments and revisions. We apologize for this referencing error, and we have manually checked all the other references to ensure they are all correct. We also noticed and corrected a lot of the broken references throughout the paper, which were due to misnamings and by a choice of splitting the main manuscript and the appendix. This unfortunately led to several broken or mispointed references. This has all been fixed in the updated version, where we have combined the appendix together with the main manuscript in a single document.
>
> ## Regarding the content and substance of the paper:
> In the updated manuscript we have moved dataset implementation details to appendix as suggested, making the main text significantly more readable.
> We also extended the results section (+2 pages) where we discuss new results that contribute to the key goals of the article:
> Scale-up results with larger models (Qwen-7B): We have scaled up our experiments to show that our results hold with larger models
> New comparison with baselines:
> Cross-task evaluation: For all our trained models, we have evaluated the cross-task performance, to control for overfitting in models, to complete Table 2. Our results now show that, expectedly, the best models at each task are those trained with RL on that task. More importantly perhaps, training with RL on a separate task does not hurt performance on other tasks, and performances remain comparable to the SFT baseline.
>
>
> More thorough ablations:
> - No-MiST ablations for multiple tasks (Tables 2, 19, 20)
> - Corruption rate sensitivity (Figure 3, Section 5.1)
>
> Results about SCS’s formulation and use as a predictive tool:
> - New study where the choice of cr=0.2 for the corruption operator is justified
> - Correlation study showing how pre-RL SCS is a good predictor for post-RL success (correlations >0.6).
>
> A case-study where we retrospectively justify the selection of Mistral-24B as the base model for the state-of-the-art reasoning model in chemistry (ether0), where we show that this is the base model with the highest SCS in its size range, despite others (e.g. Qwen-2.5-32B) performing better in most public benchmarks.
>
> ## Overclaims and evidence:
> You correctly identified that the true innovation is the diagnostic framework, not ancillary claims about task sets or data. The new version embraces this insight:
> In the updated version we have refocused the paper and contributed stronger evidence for the main claims. Our core contributions, as specified in the updated introduction, are mainly:
> 1. Quantitative diagnostics for latent solvability
> 2. MiST as a method to satisfy diagnostics
> 3. Predictive framework that enables pre-RL assessment
> 4. Empirical validation with thresholds
>
> Furthermore, we have contributed comprehensive experimental validation for these core claims.
>
> So as a summary, we:
> - Established SCS as a predictive tool with actionable thresholds: We now demonstrate that SCS can predict RL success before expensive training begins. The SCS > 1.5 threshold provides a quantitative decision criterion, transforming the contribution into a prescriptive framework with practical utility.
> - Provided independent external validation: The retrospective analysis of ether0 shows that our SCS metric correctly identifies their base model choice (Mistral-24B, SCS=2.324) as optimal among 14B-70B alternatives, despite other models like Qwen-2.5-32B performing better on standard benchmarks. This independent validation from a separate research team substantially strengthens the evidence that SCS captures genuine predictive signal.
> - Demonstrated generalizability across scales: We validated results across both 3B and 7B models. The consistent patterns show this is not a narrow finding but a generalizable principle.
> Established strong quantitative correlations: Pre-RL SCS shows robust correlation with post-RL performance (ρ > 0.6 across multiple tasks), providing empirical evidence that symbolic competence measured before RL training reliably predicts reasoning capability after RL training.
> - Finally, we have refined the discussion with a more nuanced tone, highlighting that measures like SCS can be generalized to other domains, however taking into account that a suitable corruption operator is required, something that might not be trivial in other fields.
>
>
> We hope you'll find that this revised submission addresses your concerns and delivers the "very interesting and valuable piece of work" you identified as achievable with proper validation.

---

> > ### Comment · Reviewer_udTx · 2025-11-27
> >
> > Dear Authors,
> >
> > Thank you for your detailed responses and substantial revisions. However, the current version still contains several shortcomings that I cannot overlook, which compel me to maintain my present score for now. I outline the main issues below:
> >
> > First, the overall organization of the manuscript remains rather disordered and diffuse, making it difficult for me to identify what you truly intend to highlight. The main issues include:
> >
> > 1. According to your response, you now aim to focus the paper’s main contribution on the criteria for determining the RL potential of LLMs. However, the definitions of these metrics are placed in the Preliminaries section, which typically introduces background knowledge for unfamiliar readers or summarizes foundational prior work, and usually does not contain the core innovations of the paper. The metrics then reappear only in the experiments section. As a result, the current organization still places greater emphasis on Mist rather than on the evaluation criteria.
> >
> > 2. According to your response, MiST should now be presented mainly as one approach to improving a model’s RL potential. Yet I did not find any corresponding explanation or introduction in the paper. MiST still appears as an isolated piece of content, without being connected to SCS or CCS. Chapter 4 about MiST does not even mention SCS or CCS, and the analysis of experimental results before Section 5.1 is also unrelated to RL potential. This makes the paper feel highly scattered: you first introduce the RL potential evaluation criteria, then switch to describing CPT and SFT implementations, and then move to analyzing the effects of RL training and different inference techniques(System 1 vs 2). The narrative thus appears very disjointed.
> >
> > 3. Many statements in the paper lack corresponding results. For example, the comparison between GPT-3.5 and Llama-3.1 in lines 293–294, and the comparison between the two inference techniques for the I2S task in lines 355–358.
> >
> > Second, regarding the experiments on SCS in Figure 4, they do not seem to validate the effectiveness of your proposed metrics. From Figure 4, the only clear conclusion I can draw is that models with SCS below 1 lack RL potential for chemical tasks. However, when SCS is sufficiently high, it does not appear that the SCS value can reliably predict which model will ultimately achieve better performance. Although there is some positive correlation on certain tasks, the number of distinct pre-RL SCS values is very limited (only three), and the performance overlap between models with different SCS values is very large. I suggest that to convincingly demonstrate this point, you would need RL results for more models with different pre-RL SCS values, in order to show the correlation between pre-RL SCS and post-RL performance.
> >
> > In addition, the newly added table on ether0 (which is labeled as Figure 5) does not seem to serve the purpose you state. As far as I know, the ether0 paper does not provide any evidence or argument stating something like “comparing different initial models and selecting Mistral because it performs the best.” Moreover, the table shows that the SCS score of the trained ether0 is even lower than that of Mistral before ether0 training, which seems counterintuitive. What is your interpretation of this phenomenon, and what do you believe it represents?
> >
> > These are the main issues I believe the paper still faces. I hope they will be helpful as you continue improving the quality of the manuscript.

---

### Official Review · Reviewer_uEhF · 2025-11-01

**Soundness:** 3
**Presentation:** 3
**Contribution:** 3
**Rating:** 6
**Confidence:** 4

**Summary:**

The paper introduces MiST within a multi-stage pipeline to develop chemical reasoning in LLMs. To enhance the chemical reasoning abilities, MiST applies continued pre-training on a rich 2.9B token chemistry corpus, followed by Supervised Fine-Tuning (SFT). These stages increase the latent solvability score dramatically. Crucially, the final post-training stage, employing Reinforcement Learning (RLVR), amplifies these emergent capabilities, significantly improving the performance of multiple chemistry tasks.

**Strengths:**

1. Robust Data Construction from Diverse Sources.

The model benefits from a high-quality 2.9B token corpus for continued pre-training (MiST), drawing extensively from Diverse sources such as ChemRxiv + S2ORC and PubChem synthetic data. Additional datasets including both instruction-following dataset and reasoning traces from DeepSeek-R1 are also collected for enhancing the chemical reasoning abilities.

2. Effective Multi-Stage Training Pipeline.

The training incorporates continued pre-training (MiST) for general chemistry knowledge, Supervised Fine-Tuning (SFT) using instruction datasets (like SmolInstruct) and reasoning traces distilled from DeepSeek-R1, and RLVR for specific task post-training amplification.

3. Significant Performance Improvement.

This holistic methodology raises the latent solvability of tasks and successfully amplifies capabilities, improving the performance of LLM on multiple chemistry reasoning tasks.

**Weaknesses:**

1. No experiments to show improvement in terms of SMILES generation

It's unclear whether after the extensive training with data including SMILES strings, LLM can successfully generate accurate SMILES strings while retaining good reasoning abilities now. Although the paper includes CCS and SCS scores for comparison, it is not explicit enough about the improvement in the ability to generate reasonable and correct SMILES strings.

2. No experiments to show improvement in terms of reasoning abilities

Although on 4 reasoning tasks, the trained LLM shows significant improvement in metrics compared with base model. But it is very expected and simply fine-tuning the LLM with multi-source data could also achieve similar or even higher performance based on the metrics reported in LlaSMol paper. Therefore, with only these top-1 accuracy, it is unclear whether the reasoning ability of LLM is truly improved after the extensive multi-stage training.

**Questions:**

1. is it possible to include an additional table to extensively test the ability to generate correct SMILES strings with more metrics, like whether the generated SMILES strings are grammarly correct now? At what percentage, different LLM models can output correct SMILES strings and after the training, the trained LLM can output the correct SMILES strings? Is it possible to have some specific tasks like translating SMILES correspond to the same molecule to specifically test the understanding ability of LLM towards SMILES strings after the training?

2. Can the authors show multiple reasoning examples from the trained LLM to demonstrate the improved reasoning abilities? It will also be great if more quantitative analysis based on human expert can be conducted to demonstrate the effectiveness of reasoning outputs.

---

> ### Author Response · Authors · 2025-11-16
>
> Dear Reviewer,
>
> Thank you for the constructive review, and for highlighting the strengths as well as the missing pieces of our work. Thanks to these comments we have sharpened the focus of the paper and clarified and further strengthened the evidence we provide.
>
> # To your main concerns:
>
> ## SMILES generation ability of models:
>
> We agree that we should make the explicit evaluation of SMILES generation, SCS only works as a proxy to measure this implicitly. In the original manuscript we had the tasks Iupac2Smiles, and Reaction Prediction, both of which require that the models produce valid smiles, but it is true that evaluating this mixes up both the smiles generation capability with the other required task-specific abilities.
> We would like to highlight that, after MiST, models go from essentially 0% I2S accuracy to 50-65%. We are right now working on a dedicated SMILES generation table to directly test this ability, and will be adding this to an updated manuscript in the following days.
>
> ## Improvements in reasoning abilities:
>
> We agree that the shown accuracies do not necessarily demonstrate improved reasoning, and that a deeper analysis would be needed. We have included extra reasoning traces from our models in the appendix, and have extended all our analyses to a larger scale (Qwen-7B).
>
>
> We would also like to point out that we have oriented the focus of the paper more towards answering the question “when can RL post training improve the chemical reasoning capabilities of models?” rather than directly giving a recipe for obtaining the best model. Still, we are making the reasoning aspect more concrete:
>
> - We explicitly compare System‑1 vs System‑2 prompting (direct answer vs chain‑of‑thought) for each model stage. For tasks like reaction naming (RxN), we see substantial gains from reasoning after MiST+SFT+RL; e.g. for Qwen‑7B + MiST + RL(RxN), accuracy improves from 26.4% (direct) to 63.9% (with reasoning). This suggests that reasoning traces are indeed being used to solve harder cases.
>
> A full human‑expert evaluation is beyond what we can realistically add at this point, however we are adding this as an important direction for future work, and reducing some of our claims about reasoning, where our experiments only show gains in accuracy.
>
>
> We appreciate your positive overall assessment and your suggestions for making the SMILES and reasoning aspects more explicit; we believe the added analyses and examples will make these contributions clearer and more convincing. We are happy to continue the discussion over this period to further improve the quality of our work!

---

> > ### Author Response · Authors · 2025-11-30
> >
> > Dear reviewer, it has been 2 weeks since we submitted our revised version of the manuscript along with responses to your thoughtful comments.
> > We would dearly appreciate your response for further discussion if you believe it is needed, and if you think the updated version has addressed your comments do please consider raising the score accordingly.
> >
> > All the best,
> > The authors.

---

### Author Response · Authors · 2025-11-16

We thank all reviewers for their constructive and very detailed feedback. The updated manuscript has substantially improved over the time after the first submission, and thanks to the reviews. We have now implemented several revisions to address the main concerns from the reviewers and ultimately give a better and more focused shape to the manuscript.

The paper now has a stronger focus on diagnostic metrics as predictive tools, that can help assess the readiness of base models for RL training on chemical tasks. MiST is now positioned as a method that can help satisfy such readiness, rather than a core contribution itself. In this vein, we have updated the manuscript with several more experiments that directly show the value of measures like SCS at predicting future success, and others that help clarify the construction of the metric, justifying decisions that at first seemed ad-hoc. Finally, we conducted a case study that further showcases the predictive power of this approach.

Here we summarize some of the main additions and fixes we have introduced in this new version of the manuscript. The changes have been highlighted in blue in the new version to facilitate revision.

- Added Section 5.1: Formulation and validation of SCS:
We analyze the effect of varying the corruption rate in the formulation of SCS, and find that 0.2 is enough to cause differentiation between the models.
- Added Section 5.2: SCS as a predictive framework:
Correlation analysis between pre-RL SCS against post-RL performance. We show ρ >0.6 cross-task performances, and show a clear success signal post-RL when evaluated in-task (Figure 4). We also include a case-study, where we determine the best large base model for RL training from a pool of open-access models. The model (Mistral-24B) matches the one used as base for ether0 (Narayanan 2025), retrospectively confirming the optimality of this choice, despite it having lower benchmark scores than similarly sized or even larger LLMs.
- Multi-scale validation: All experiments have now also been scaled to 7B, showing consistent observations across scales.
- Additional baselines: We now include ChemLLM-7B as a new baseline in Table 2, showing even our 3B models overperform it.

Additionally we have improved the presentation and organization of the paper, combining the manuscript and appendix into a single document, which addresses the broken appendix references. Dataset processing details have been mostly moved to the appendix, and Sections 4 and 5 have been largely revised to match the flow of the first Sections. Other formatting issues as noted by reviewers have also been fixed, as well as improved mathematical notation and revised citations.
As a summary, the new version has shifted the tone into a prescriptive framework where we show that SCS and SCS-like measures can predict success of RL on specific domains, chemistry in our case, with actionable thresholds that can be computed on small scale but that translate into much larger models. MiST finds a role as an enabler of such readiness, to optimize models against SCS while preventing over-fitting to scientific notation.
We believe the work has substantially improved and we thank the reviewers again for their constructive comments and in-depth revisions. We hope the strengths the reviewers had originally identified are now even stronger in the new paper, and we are open to further enriching discussion and revision!

---

### Note · Program_Chairs · 2026-01-17
**Submission Desk Rejected by Program Chairs**

The following references in this submission do not refer to real documents and/or have major errors in bibliographic information:

 Łukasz Maziarka, Krzysztof Rataj, Tomasz Danel, Piotr Warchoł, and Stanisław Jastrzębski. ChemBERTa-2: Large-scale self-supervised pre-training for molecules. arXiv preprint arXiv:2309.12948, 2023.